# Neural Collapse by Design: Learning Class Prototypes on the Hypersphere

**Panagiotis Koromilas** [1]   **Theodoros Giannakopoulos** [2]   **Mihalis A. Nicolaou** [3 4]   **Yannis Panagakis** [1 5]

## Abstract

Supervised classification has a theoretical optimum, Neural Collapse (NC), yet neither of its two dominant paradigms reaches it in practice. Cross entropy (CE) leaves radial degrees of freedom unconstrained and converges to a degenerate geometry, while supervised contrastive learning (SCL) drives features toward NC during pretraining but discards this structure in a post hoc linear probing phase. We show that both paradigms are different appearances of the same method that contrasts prototypes on the unit hypersphere, and that closing the gap requires fixing each at its point of failure. From the CE side, we propose NTCE and NONL, two normalized losses that import contrastive optimization's missing ingredients into classifier learning: a large effective negative set and decoupled alignment and uniformity terms. From the SCL side, we prove that SCL's objective already optimizes throughout training for a principled classifier whose weights are the class mean embeddings, making linear probing both redundant and harmful. Empirically, on four benchmarks including ImageNet-1K, NTCE and NONL surpass CE accuracy, closely approximate NC ($\geq 95\%$), and match CE's converged NC on 4/5 metrics in under $7.5\%$ of its iterations, while SCL with fixed prototypes matches linear probing without the hours-long classifier training phase. The learned geometry yields $+5.5\%$ mean relative improvement in transfer learning, up to $+8.7\%$ under severe class imbalance, and improved robustness to corruptions on ImageNet-C. Our work recasts supervised learning as prototype learning on the hypersphere, with NC reached *by design*. Code: https://github.com/pakoromilas/nc_by_design

[1]University of Athens [2]NSCR Demokritos [3]University of Cyprus [4]The Cyprus Institute [5]Archimedes AI/Athena Research Center. Correspondence to: Panagiotis Koromilas <pakoromilas@di.uoa.gr>.

*Proceedings of the $43^{rd}$ International Conference on Machine Learning*, Seoul, South Korea. PMLR 306, 2026. Copyright 2026 by the author(s).

## 1. Introduction

Despite theoretical proofs that Neural Collapse (NC) is the global optimum of supervised learning objectives (Graf et al., 2021; Lu & Steinerberger, 2022; Zhou et al., 2022a), standard supervised pipelines rarely attain it in practice. This failure is particularly striking because *NC delivers precisely the properties we seek*: when neural networks do approach this geometric configuration, where within class representations collapse to their means, class means form an equiangular tight frame (ETF), and classifier weights align with these prototypes, they demonstrate improved generalization (Bartlett et al., 2017; Neyshabur et al., 2018; Papyan et al., 2020), adversarial robustness (Ding et al., 2020; Fawzi et al., 2016), enhanced transfer learning (Galanti et al., 2021; Khosla et al., 2020), and converge toward max margin classifiers (Soudry et al., 2018) with stronger robustness guarantees (Hein & Andriushchenko, 2017). *If NC is provably optimal and empirically beneficial, why do standard supervised pipelines consistently fail to achieve it?*

We identify the core issue as *unconstrained radial degrees of freedom*. Cross entropy (CE) optimization allows features and weights to be jointly rescaled without changing predictions (Soudry et al., 2018), leaving radial directions underconstrained and preventing convergence to a unique geometry. While explicit regularization of features, weights, and biases may resolve this (Zhu et al., 2021), it introduces multiple hyperparameters that complicate practical adoption. A principled solution is to eliminate radial freedom entirely by constraining optimization to the unit hypersphere, where NC becomes the *unique* global optimum (Yaras et al., 2022). This is exactly what *normalized softmax* losses such as NormFace (Wang et al., 2017) do: by normalizing both features and classifier weights, they project classification onto the hypersphere and reformulate it as angular similarity between data representations and learnable class prototypes.

Yet CE is not the only supervised paradigm that fails to reach NC. Supervised contrastive learning (SCL) (Khosla et al., 2020) does drive features toward the NC geometry during pretraining, mapping them onto the unit hypersphere through a projection head and producing within class collapse and ETF aligned class means (Graf et al., 2021). The standard pipeline, however, then lifts to a different representation space by discarding the projection head and trains

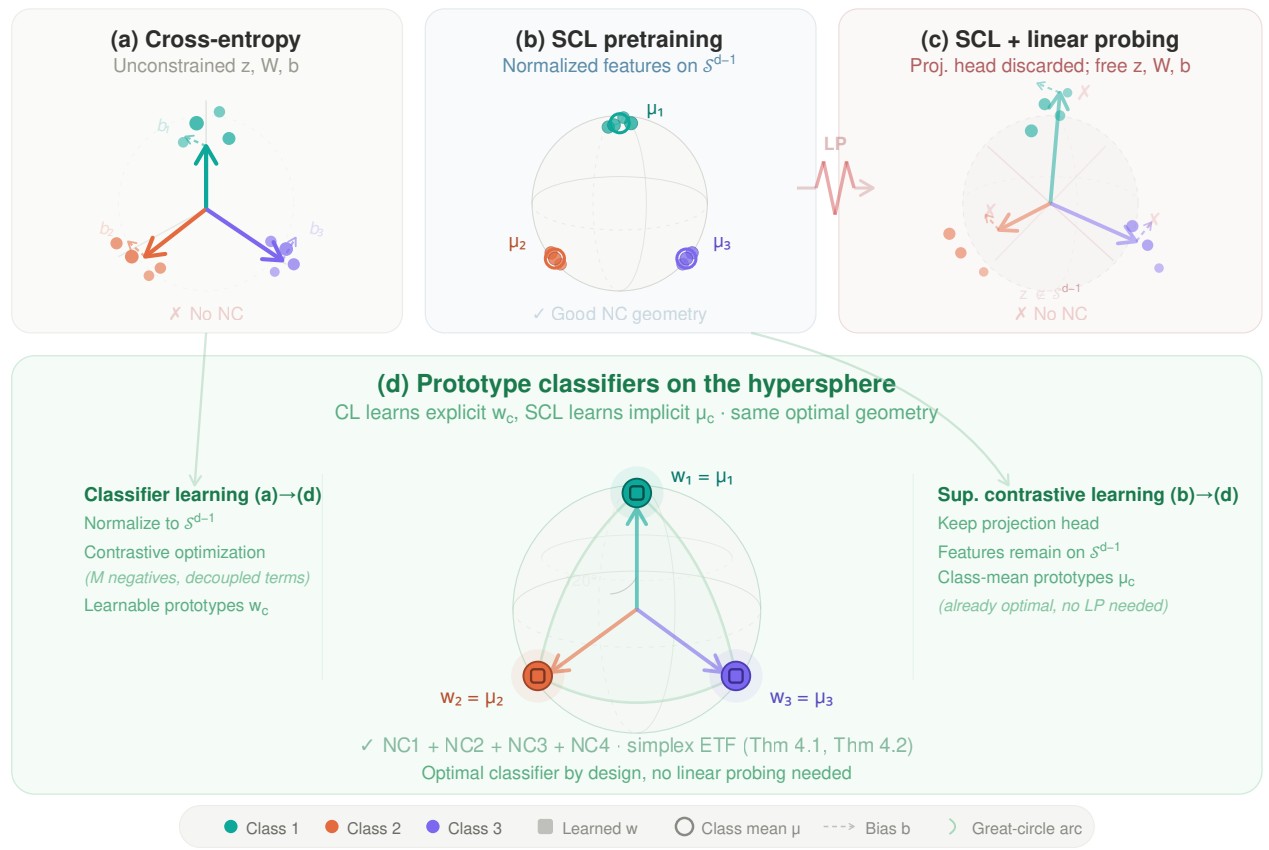

Figure 1. **Supervised learning as learning class prototypes on the hypersphere. (a)** Cross-entropy with unconstrained features $\mathbf{z}$, weights $\mathbf{W}$, and biases $\mathbf{b}$ leaves radial degrees of freedom free, preventing convergence to NC. **(b)** SCL pretraining maps features onto $\mathcal{S}^{d-1}$ via a projection head, producing representations that approach within-class collapse (NC1) and maximal between-class separation (NC2). **(c)** Standard practice discards the projection head and trains a linear probe on unconstrained $\mathbf{z}$, reintroducing free $\|\mathbf{w}\|$ and $\mathbf{b}$ that destroy the NC geometry learned during pretraining. **(d)** We show that both paradigms learn class prototypes on the hypersphere, converging to the same simplex ETF. From classifier learning (CL): normalizing to $\mathcal{S}^{d-1}$ and applying contrastive optimization (NTCE/NONL) yields learnable prototypes $\hat{\mathbf{w}}_c$ that converge to class means (Theorem 4.1). From SCL: the class-mean prototypes $\hat{\boldsymbol{\mu}}_c$ are already the optimal classifier throughout training, making linear probing unnecessary (Theorem 4.2). Both paths achieve NC1–NC4 in theory and closely approximate it in practice, with $\hat{\mathbf{w}}_c = \hat{\boldsymbol{\mu}}_c$ at the global optimum.

a linear classifier on the unnormalized encoder representations with free weights and biases, reintroducing exactly the radial and bias pathologies that broke CE in the first place. The two paradigms therefore fail for opposite reasons: *CE never builds the geometry, while SCL approximates it during pretraining only to discard it during linear probing*.

These complementary diagnoses converge on a single conceptual lens: *prototype contrast on the unit hypersphere* (Figure 1). Once Classifier learning (CL) adopts normalized softmax, it optimizes angular similarity between normalized features and *explicit* class weight vectors that serve as prototypes. We prove that SCL, on the other side, optimizes angular similarity among normalized instances using *implicit* class mean embeddings as prototypes. CL and SCL are therefore *different appearances of the same method*, differing only in whether the prototype is parameterized or

emergent, and both can reach the same simplex ETF.

Despite this shared geometric foundation, normalized softmax recasts CE in contrastive form but does not yet bring the corresponding optimization with it: it contrasts against only $K$ class prototypes (He et al., 2020), giving a small effective negative set, and couples positive and negative similarity terms through a shared normalization (Yeh et al., 2022). We close the remaining gaps to NC by applying *proper contrastive optimization* and making the following **five contributions**:

**C1.** We **unify normalized softmax and SCL** under a single geometric framework, revealing both as *prototype contrast methods on the unit hypersphere* that differ only in whether prototypes are explicit (learned weights) or implicit (class means). This framework explains why both can achieve NC in practice while standard CE cannot.

**C2.** We propose **two supervised objectives** that overcome existing computational limitations. **NTCE** (Normalized Temperature scaled Cross Entropy) increases the effective number of negatives from $K$ classes to $M$ batch samples, strengthening inter class separation. **NONL** (Negatives Only Normalization Loss) eliminates interference between intra class alignment and inter class repulsion by normalizing only over negatives, accelerating NC convergence.

**C3.** We prove that *the SCL objective already optimizes for an optimal prototype classifier throughout pretraining*, **eliminating the need for linear probing**. The class-mean embeddings learned by SCL form the principled SCL classifier regardless of whether NC is attained.

**C4.** We validate our approach across four benchmarks including ImageNet-1K. NTCE and NONL achieve $\geq 95\%$ *on NC metrics* while *surpassing standard CE accuracy*, and match CE's NC metrics with *substantially fewer training iterations*. Our prototype classifier maintains SCL's accuracy while eliminating hours of linear probing computation, a significant *practical saving for large scale deployments*.

**C5.** We empirically demonstrate that the representations learned by our objectives translate into **practical benefits**, yielding improved performance on *transfer learning* (+5.5% mean relative improvement), *long tailed classification* (up to +8.7% relative improvement), and *robustness* (lower mCE).

These results suggest a **fundamental shift** in how supervised learning should be understood: not as unconstrained optimization in Euclidean space, but as *prototype based classification on the hypersphere*.

## 2. Related Work

**Neural Collapse.** Neural Collapse (NC) describes a limiting geometry in which within-class features collapse to their means (NC1), class means form a centered simplex ETF (NC2), classifier weights align with the means (NC3), and biases collapse (NC4) (Papyan et al., 2020). Variants of this structure characterize global minimizers for several objectives and modeling assumptions, including MSE (Han et al., 2022; Zhou et al., 2022a), cross-entropy (CE) (Lu & Steinerberger, 2022), supervised contrastive learning (SCL) (Graf et al., 2021), and CE variants such as label smoothing and focal loss (Zhou et al., 2022b). In finite training, however, standard CE with weight decay often fails to realize the optimal geometry: the loss is *scale-noncoercive* and can be driven toward zero by inflating logit magnitudes without improving angular structure (Albert & Anderson, 1984; Soudry et al., 2018). Class imbalance further distorts the ETF and slows convergence (Hong & Ling, 2024; Thrampoulidis et al., 2022); free bias terms obstruct NC4 and can exacerbate miscalibration unless controlled (e.g., logit adjustment) (Menon et al., 2021). While simultane-ously penalizing features, weights, and biases can restore coercivity and yield NC in principle (Zhou et al., 2022a; Zhu et al., 2021), tuning multiple regularizers is brittle. *We show that contrasting instances against class prototypes on the hypersphere operationalizes NC in practice.*

**Learning on the hypersphere.** Constraining radial freedom is a principled route to NC. When both features and classifier lie on the unit hypersphere, CE over the product of spheres exhibits a benign strict-saddle landscape whose minima realize perfect NC (Yaras et al., 2022). Related evidence appears in contrastive objectives: SCL yields within-class collapse and simplex class means (Graf et al., 2021), while in self-supervised contrastive learning batch-level optima form a simplex ETF (Koromilas et al., 2024). A long line of face-recognition work, including SphereFace, CosFace, ArcFace, and NormFace (Deng et al., 2019; Liu et al., 2017; Wang et al., 2017; 2018), operationalizes direction-only discrimination by using angular/cosine margins. *We unify these approaches by showing that both normalized softmax and SCL perform prototype contrast on the hypersphere.* Building on this bridge, we extend normalized softmax with NTCE/NONL to import desirable properties.

**Prototype-based classification and ETF classifiers.** Prototype methods classify via distances to learned representatives (Snell et al., 2017). Motivated by NC, several works fix or guide the classifier toward ETF-like prototypes and learn only the encoder, for example by (i) fixing a simplex ETF head and training the backbone (ETF+DR) (Yang et al., 2022), (ii) using hyperspherical prototype networks (Mettes et al., 2019), or (iii) constructing equiangular basis vectors (EBVs) (Shen et al., 2023). Other approaches enforce (non-negative) orthogonality (Kim & Kim, 2024) or guide the classifier toward the nearest ETF via a Riemannian inner optimization (Markou et al., 2024). Recently NC structure has been exploited in a teacher–student setting (Zhang et al., 2025): given a trained teacher that already exhibits NC, they compute *teacher* class centroids and use them as an NC3-inspired classifier for the *student*. *Our perspective is that CL and SCL already operate with prototypes: we modify the objectives to realize NC in practice, and we show that SCL's class-mean prototypes form an effective classifier, making linear probing unnecessary.*

## 3. Preliminaries

**Notation.** Scalars are denoted by lowercase letters $u$, vectors by lowercase bold letters $\boldsymbol{u}$, and matrices by uppercase bold letters $\boldsymbol{U}$. Sets are represented by uppercase caligraphic letters $\mathcal{U}$. Individual elements are accessed using subscript notation: $u_i$ for the $i$-th element of vector $\boldsymbol{u}$ and $U_{i,j}$ for the element at row $i$ and column $j$ of matrix $\boldsymbol{U}$. To denote vertical (row-wise) concatenation of matrices $\mathbf{X}$ and $\mathbf{Y}$, we use $[\mathbf{X}; \mathbf{Y}]$. We denote normalized vectors with

$\hat{\boldsymbol{u}}_j = \boldsymbol{u}_j/\|\boldsymbol{u}_j\|$.

## 3.1. Learning Paradigms

**Classifier Learning with Cross-Entropy.** The cross-entropy loss is the standard Classifier Learning (CL) objective, optimizing representations and classifier weights simultaneously. An encoder $f_{\boldsymbol{\theta}} : \mathcal{X} \to \mathcal{Z}$, parameterized by $\boldsymbol{\theta} \in \Theta$, maps an input $\mathbf{x} \in \mathcal{X}$ to its representation $\mathbf{z} = f_{\boldsymbol{\theta}}(\mathbf{x}) \in \mathcal{Z}$. For a $K$-class task, $y_i$ denotes the class assignment of sample $\mathbf{x}_i$. A linear classifier is placed on top of the encoder, with weight matrix $\mathbf{W} \in \mathbb{R}^{K \times h}$ and bias $\mathbf{b} \in \mathbb{R}^K$, where $h$ is the embedding dimension. For a mini-batch of $M$ samples with $\{\mathbf{z}_i\}_{i=1}^M$, the cross-entropy loss is defined as

$$\mathcal{L}_{\text{CE}}(\mathbf{Z}, \mathbf{W}) = \frac{1}{M} \sum_{i=1}^{M} - \log \left( \frac{e^{\mathbf{z}_i^\top \mathbf{w}_{y_i} + b_{y_i}}}{\sum_{j=1}^{K} e^{\mathbf{z}_i^\top \mathbf{w}_j + b_j}} \right), \quad (1)$$

where $\mathbf{w}_j$ denotes the $j$-th row of $\mathbf{W}$ and $b_j$ the $j$-th component of $\mathbf{b}$.

**Supervised Contrastive Learning.** *Supervised Contrastive Learning (SCL)* takes a seemingly different direction: it learns representations by exploiting similarities between instances to learn class-invariant representations. Building on our notation, the contrastive framework augments the encoder $f_{\boldsymbol{\theta}} : \mathcal{X} \to \mathcal{Z}$ with a projection head $g_{\boldsymbol{\phi}} : \mathcal{Z} \to \mathcal{U}$, parameterized by $\boldsymbol{\phi} \in \Phi$, which maps representations onto the unit hypersphere, $\mathcal{U} = \mathbb{S}^{d-1} = \{\boldsymbol{u} \in \mathbb{R}^d \mid \|\boldsymbol{u}\| = 1\}$. We denote the projected representations as $\boldsymbol{u}, \boldsymbol{v} \in \mathcal{U}$, where $\boldsymbol{u}_i$ comes from instance $\boldsymbol{x}_i$ and $\boldsymbol{v}_i$ from its alternative view produced via augmentation, a typical process in contrastive learning.

For SCL the objective is to pull together positive pairs while pushing apart negative pairs in the projection space. Typically alternative views of the same data point that originate from augmentation are considered as new data points, *i.e.* $\boldsymbol{A} = [\boldsymbol{U}; \boldsymbol{V}]$, and the supervised contrastive loss becomes:

$$\mathcal{L}_{\text{SCL}}(\boldsymbol{A}) = \frac{1}{2M} \sum_{i=1}^{2M} \frac{-1}{|\mathcal{C}(i)|} \sum_{l \in \mathcal{C}(i)} \log \left( \frac{e^{\boldsymbol{a}_i^\top \boldsymbol{a}_l / \tau}}{\sum_{\substack{j=1 \\ j \neq i}}^{2M} e^{\boldsymbol{a}_i^\top \boldsymbol{a}_j / \tau}} \right),$$
$$(2)$$

where $\mathcal{C}(i)$ denotes the set of indices corresponding to positive examples sharing the same class as $\boldsymbol{x}_i$ and $\tau > 0$ is a temperature parameter that controls the concentration of the distribution.

A crucial distinction emerges post-training: while learning with cross-entropy directly produces a classifier, contrastive learning requires an additional step. After optimizing Equation (2), the projection head is discarded and a linear classifier $\boldsymbol{W}, \boldsymbol{b}$ is trained on the frozen encoder representations $\boldsymbol{z}$ using Equation (1), a process known as **linear probing**.

## 3.2. Neural Collapse (NC).

*Neural Collapse* Papyan et al. (2020) is the late-training regime (on balanced data) where last-layer features and the linear classifier converge to a highly structured limit. Let $\boldsymbol{z}_i = f(\boldsymbol{x}_i) \in \mathbb{R}^h$, class means $\boldsymbol{\mu}_c = \frac{1}{n_c} \sum_{i:y_i=c} \boldsymbol{z}_i$, weights $\boldsymbol{w}_c$, and bias $\boldsymbol{b}$. NC holds when, up to common scalings:

(NC1) **Within-class collapse:** $\boldsymbol{z}_i = \boldsymbol{\mu}_{y_i}$ for all $i$.

(NC2) **Simplex ETF of class means:** the centered means $\tilde{\boldsymbol{\mu}}_c = \boldsymbol{\mu}_c - \frac{1}{K} \sum_{k=1}^{K} \boldsymbol{\mu}_k$ have equal norms and equal pairwise angles so the means span a centered $(K-1)$-simplex ETF.

(NC3) **Alignment of Class Representation and Classifier:** classifier columns align with the class means, $\boldsymbol{w}_c \parallel \boldsymbol{\mu}_c$ (there exists $\gamma > 0$ with $\boldsymbol{w}_c = \gamma \boldsymbol{\mu}_c$).

(NC4) **Bias collapse:** $\boldsymbol{b} = \beta \mathbf{1}$ for some scalar $\beta$.

Under NC, the decision rule reduces to nearest-class-mean classification. We assume balanced classes and $h \geq K - 1$ so a centered simplex ETF is feasible (Lu & Steinerberger, 2022).

**Practical Challenges in reaching Neural Collapse** Neural Collapse (NC) is now well documented in deep nets (Papyan et al., 2020) and characterizes global minima of balanced cross-entropy (Lu & Steinerberger, 2022). However standard pipelines does not enforce it in practice. For the typical paradigm of cross-entropy and classifier weight decay, the objective admits an *unbounded rescaling direction*: shrinking the classifier while amplifying features leaves logits unchanged, reduces the penalty, and drives the loss toward zero without achieving NC (Albert & Anderson, 1984; Soudry et al., 2018). It is shown by Zhu et al. (2021) that a well-posed objective arises when all radial degrees of freedom are constrained by penalizing weights, features, and biases simultaneously (Zhu et al., 2021). However this is practically brittle due to multiple regularizers to tune.

Supervised contrastive training on the other hand can drive representations toward NC geometry (Graf et al., 2021). However, the subsequent *linear probing* step typically fits a softmax classifier with cross-entropy on *frozen* features, allowing free weight magnitudes and biases. This reintroduces the same scale and bias pathologies as cross-entropy even when training has already reached an NC.

## 4. Supervised Learning on the Hypersphere

In this section we bridge classifier learning and supervised contrastive learning under a common viewpoint. Our approach uses similarity-based optimization while eliminating

radial degrees of freedom by constraining both feature and classifier norms to the hypersphere. This constraint transforms the optimization landscape into a benign geometry where all critical points become global optima (Yaras et al., 2022), enabling direct convergence to NC.

### 4.1. Revisiting Cross Entropy: Contrasting Class Prototypes to Instances

The weight matrix of the final linear classifier in CL methods can be expressed as $\boldsymbol{W} = [\boldsymbol{w}_1; \boldsymbol{w}_2; \ldots; \boldsymbol{w}_K] \in \mathbb{R}^{K \times h}$, where each $\boldsymbol{w}_c$ represents a learnable class prototype. This formulation reveals an important insight: we can treat the classifier weights as *learnable prototypes* that evolve through gradient descent to capture class-specific geometric structures. Building on this we design objectives that use such prototypes to arrive at the optimal NC geometry.

**Normalized Softmax Losses.** Standard cross-entropy and contrastive learning represent two seemingly distinct paradigms: the former discriminates through learned magnitudes and biases in unconstrained space, while the latter operates purely on angular similarities on the hypersphere. This fundamental difference leads to a critical inefficiency: while both methods theoretically converge to neural collapse configurations, cross-entropy introduces unnecessary radial degrees of freedom that slow convergence to this optimal geometry (Yaras et al., 2022; Zhu et al., 2021).

Normalized softmax losses resolve this inefficiency by reformulating cross-entropy as a pure geometric objective. NormFace (Wang et al., 2017), a prominent example, achieves this through three coordinated modifications: (i) eliminating biases that merely translate decision boundaries without encoding semantic structure, (ii) projecting representations onto the hypersphere to focus exclusively on angular geometry, and (iii) introducing temperature scaling to control concentration of the softmax distribution. Formally, with $\boldsymbol{u}_i = \boldsymbol{z}_i/\|\boldsymbol{z}_i\|_2$ as the normalized representation and $\hat{\boldsymbol{w}}_j = \boldsymbol{w}_j/\|\boldsymbol{w}_j\|_2$ as the normalized classifier weight for class $j$, NormFace minimizes:

$$L_{\text{NormFace}}(\boldsymbol{U}, \boldsymbol{W}) = -\frac{1}{M} \sum_{i=1}^{M} \log \left( \frac{e^{\boldsymbol{u}_i^\top \hat{\boldsymbol{w}}_{y_i}/\tau}}{\sum_{j=1}^{K} e^{\boldsymbol{u}_i^\top \hat{\boldsymbol{w}}_j/\tau}} \right). \tag{3}$$

This reformulation transforms classification into contrastive learning between data instances and learnable class prototypes while maintaining CE's computational efficiency.

**Normalized Temperature-scaled Cross Entropy (NTCE).** When viewing CL from a contrastive learning perspective through NormFace we encounter an inherent limitation of cross entropy: the number of negatives in the objective is limited to $K$, the number of class prototypes. Contrastive objectives need very large numbers of negatives in order to

converge (He et al., 2020), since fewer negatives provide a worse estimate of the expectation of the actual contrastive objective (Koromilas et al., 2024).

By inverting the contrastive direction from instance-to-class to class-to-instance discrimination we address this limitation through the Normalized Temperature-scaled Cross Entropy (NTCE). This modification fundamentally alters the learning dynamics: rather than each instance contrasting against $K$ class prototypes, each class prototype now contrasts against $M$ batch representations.

The key insight underlying NTCE is that *class prototypes themselves can serve as anchors* in the contrastive formulation. By anchoring on the class weight vector corresponding to each instance's ground-truth label and contrasting it against all batch representations, we dramatically expand the negative sampling space. Formally, NTCE takes the form:

$$L_{\text{NTCE}}(\boldsymbol{U}, \boldsymbol{W}) = \frac{1}{M} \sum_{i=1}^{M} - \log \left( \frac{e^{\hat{\boldsymbol{w}}_{y_i}^\top \boldsymbol{u}_i/\tau}}{\sum_{j=1}^{M} e^{\hat{\boldsymbol{w}}_{y_i}^\top \boldsymbol{u}_j/\tau}} \right), \tag{4}$$

where $\hat{\boldsymbol{w}}_{y_i}$ serves as the anchor for instance $i$, and critically, the denominator sums over all $M$ instances in the batch rather than over $K$ classes.

**Negatives Only Normalization Loss (NONL).** NTCE adds enhanced negative sampling on top of NormFace to directly transfer the principles of contrastive learning to cross entropy training. However, it also inherits a fundamental drawback of popular contrastive objectives that compromises its optimization dynamics. The denominator in Equation (4) indiscriminately aggregates all instances sharing the same class anchor. That is, the denominator (also known as the uniformity term) is optimized when instances of the same class have maximum distance (Wang & Isola, 2020), which contradicts the optimality of the numerator (the alignment term). More specifically, positive pairs explicitly appear as negative samples in the normalization term, generating gradients that actively repel instances from their own class prototype. When instance $i$ and instance $j$ share class $y_i = y_j$, the term $e^{\hat{\boldsymbol{w}}_{y_i}^\top \boldsymbol{u}_j/\tau}$ in the denominator produces gradients that decrease $\hat{\boldsymbol{w}}_{y_i}^\top \boldsymbol{u}_j$, directly opposing the alignment objective. This is a known behavior called *alignment-uniformity coupling* (Koromilas et al., 2024; Yeh et al., 2022).

To resolve this conflict we introduce the Negatives-Only Normalization Loss (NONL), which explicitly excludes same-class instances from the denominator:

$$L_{\text{NONL}}(\boldsymbol{U}, \boldsymbol{W}) = \frac{1}{M} \sum_{i=1}^{M} - \log \left( \frac{e^{\hat{\boldsymbol{w}}_{y_i}^\top \boldsymbol{u}_i/\tau}}{\sum_{\substack{j=1 \\ j \notin \mathcal{C}(i)}}^{M} e^{\hat{\boldsymbol{w}}_{y_i}^\top \boldsymbol{u}_j/\tau}} \right). \tag{5}$$

## 4.2. Revisiting Supervised Contrastive Learning: Contrasting Mean-Class Prototypes to Instances

Standard SCL pipelines train an encoder with Equation (2) on the unit hypersphere, then discard the projection head and fit a linear classifier with cross-entropy on the unnormalized encoder representations. This phase, known as linear probing, introduces **unnecessary degrees of freedom** that disrupt the geometric optimality achieved during pretraining: (i) **geometric mismatch**: SCL features live on the hypersphere with collapsed, ETF-structured class means; linear probing operates in unconstrained Euclidean space, allowing weight rescaling and bias shifts that break classifier-feature alignment (Soudry et al., 2018), and (ii) **computational redundancy**: it is a separate training phase, typically of $T$ epochs over the whole dataset.

Prior work (Graf et al., 2021) shows that SCL minimizers exhibit within-class collapse and a centered simplex ETF of class means. If both SCL and a hypothetical classifier converge to the same NC geometry, then the class-mean prototypes $\hat{\boldsymbol{\mu}}_c$ themselves already satisfy NC3 by construction. A natural alternative to linear probing is therefore **Fixed Prototypes (FP)**: keep the projection head (where SCL features are optimized), and set the classifier weights directly to the prototypes, $\boldsymbol{w}_c = \hat{\boldsymbol{\mu}}_c$, with no additional training.

Despite this being a straightforward idea from the theoretical results of Graf et al. (2021), there are several reasons to expect this to not work. Exact NC is never reached in practice, where SCL trained models do not realize the ETF (see Table 3). Therefore, in the non-optimal case we have no *a priori* reason to believe class-mean prototypes are the classifier this objective drives toward. Theorem 4.2 answers whether FP is a proper classifier for SCL.

## 4.3. A Unified View of Supervised Learning

We first show in Theorem 4.1 that, in the balanced UFM/LPM setting (Tirer & Bruna, 2022; Yaras et al., 2022), the three normalized losses (NormFace, NTCE, and NONL) are globally optimized by Neural Collapse (NC) geometry.

**Theorem 4.1** (**Neural Collapse optimality of normalized losses**). *In the balanced UFM/LPM setting above with $d \geq K$, every global minimizer of $L_{\mathrm{NF}}$, $L_{\mathrm{NTCE}}$, and $L_{\mathrm{NONL}}$ satisfies NC1–NC3 (within-class collapse, simplex ETF class means, and classifier–feature alignment), up to a global rotation and permutation of class labels.*

We now return to the FP classifier from Section 4.2. We follow Equation (2) to treat alternative views produced by data augmentation as distinct samples, $\boldsymbol{A} = [\boldsymbol{U}; \boldsymbol{V}]$. Let $\mathcal{B}_c = \{j \in [2M] : y_j = c\}$ denote the within-batch index set for class $c$, $n_c = |\mathcal{B}_c|$, and $\hat{\boldsymbol{\mu}}_c = \frac{1}{n_c} \sum_{j \in \mathcal{B}_c} \boldsymbol{a}_j$ the corresponding batch prototype (class mean). We define the

*prototype loss, $L_{\mathrm{proto}}(\boldsymbol{A}) =$*

$$-\frac{1}{2M} \sum_{i=1}^{2M} \log \left( \frac{e^{\boldsymbol{a}_i^\top \hat{\boldsymbol{\mu}}_{y_i}/\tau}}{-e^{\boldsymbol{a}_i^\top \hat{\boldsymbol{\mu}}_{y_i}/\tau} + \sum_{c=1}^{K} n_c \cdot e^{\boldsymbol{a}_i^\top \hat{\boldsymbol{\mu}}_c/\tau}} \right),$$
(6)

where the numerator encourages alignment with the correct class prototype, while the denominator includes both positive and negative prototypes weighted by their batch frequencies $n_c$. Theorem 4.2 connects the optima of this loss to the ones of SCL. The proof can be found in Section B.

**Theorem 4.2** (**Equivalence of SCL and prototype–softmax minimizers**). *For unit-norm representations and balanced labels the supervised contrastive loss $L_{SCL}$ and the prototype loss $L_{proto}$ in Equation (6) share the same set of global minimizers (up to rotation and label permutation). In particular, at every global minimizer the representations exhibit in-class collapse and the class means form a centered simplex ETF.*

Theorem 4.2 closes the gaps raised in Section 4.2. It proves the minimizer-set equality $\arg\min L_{\mathrm{SCL}} = \arg\min L_{\mathrm{proto}}$, meaning that SCL's objective is equivalent to optimizing the prototype-softmax classifier of Equation (6). The class-mean prototypes are therefore not a post-hoc choice but the solution the SCL objective drives toward across the entire optimization trajectory, not only at the unreachable exact minimum. This enables us to discard linear probing by setting $\boldsymbol{w}_c = \hat{\boldsymbol{\mu}}_c$ directly. In our FP method, classifier weights are the learned class means, taken from the projection-head output where SCL features actually live, with no additional magnitude or bias. We show empirically (Section 5) that FP matches linear probing accuracy while eliminating a separate training phase. We discuss the relation to Graf et al. (2021) and to Kini et al. (2024), who analyze SCL minimizers under specific architectural assumptions, in Sections D.1 and D.4.

**The unified view.** The $n_c$ weighting in the denominator of Equation (6) captures the effect of using multiple negative instances, matching the structure of Equation (4). When discarding the $n_c$ weights, this loss reduces to Equation (3), establishing a direct **correspondence between the prototype weights and class means**. The optimal solution of Equation (3) holds when $\boldsymbol{w}_c = \hat{\boldsymbol{\mu}}_c$ (Yaras et al., 2022), and our results above extend this correspondence to NTCE and NONL on the CL side and to SCL on the contrastive side. Classifier learning with normalized losses and supervised contrastive learning are therefore two appearances of the same object: prototype contrast on the unit hypersphere. The two paradigms differ only in whether the prototype is parameterized (the learnable $\boldsymbol{w}_c$ in CL) or emergent (the class mean $\hat{\boldsymbol{\mu}}_c$ in SCL). Under this view, supervised learning becomes a question of *directions*, not magnitudes, with a single optimal geometry, the simplex ETF, reachable from both sides.

*Table 1.* **Classifier Learning Performance Comparison.** green: overall best per dataset.

| Loss | CIFAR-10 | CIFAR-100 | ImageNet-100 | ImageNet-1K |
|------|----------|-----------|--------------|-------------|
| CE | 94.6 | 72.1 | 84.4 | 75.4 |
| ETF + DR | 94.4 | 72.1 | 84.5 | 75.4 |
| NORMFACE | 94.8 | 72.4 | 84.4 | 76.4 |
| NTCE (ours) | 94.7 | 72.9 | 84.7 | **76.7** |
| NONL (ours) | **94.9** | **73.6** | **84.9** | 76.5 |

*Table 2.* **Supervised Contrastive Learning Performance Comparison.** Green: best per dataset. $T$: epochs; LP: Linear Probing; NLP: Normalized Linear Probing; FP: Fixed Prototypes.

| Method | Loss | Passes | CIFAR-10 | CIFAR-100 | IN-100 | IN-1K |
|--------|------|--------|----------|-----------|--------|-------|
| LP | SCL | $T \times N$ | **95.0** | **73.9** | 84.8 | **75.1** |
| NLP | SCL | $T \times N$ | 94.9 | 73.6 | 84.8 | **75.1** |
| FP (ours) | SCL | $N$ | **95.0** | **73.9** | **86.8** | **75.1** |

# 5. Empirical Validation and Discussion

In this section we empirically validate our methods against cross-entropy (CE), ETF + DR (Yang et al., 2022), and NormFace(Wang et al., 2017) for Classifier Learning paradigms and supervised contrastive learning (SCL), evaluating: (i) classification accuracy, (ii) proximity to neural collapse geometry, and (iii) NC convergence speed. Experiments are conducted on four standard datasets: *CIFAR10, CIFAR100, ImageNet-100, and ImageNet1K*, following common representation learning benchmarking practices (Khosla et al., 2020; Markou et al., 2024; Wang et al., 2021; Yeh et al., 2022). We use ResNet50 for ImageNet datasets and ResNet18 for CIFAR. Implementation details are provided in Section E.

## 5.1. Classification Performance

**Classifier Learning Methods.** As can be inferred from Table 1, normalized losses *outperform cross-entropy* (CE) in all cases, while also our losses further outperform NormFace. NONL achieves the strongest gains on datasets with few (10) to medium (100) number of classes while it has the second best score on ImageNet-1K. Here we have to note that ImageNet-1K, our objectives exhibit the typical behavior of contrastive-style objectives: they benefit from larger batch sizes, whereas using smaller batches leads to degraded performance due to insufficient in-batch negatives (see Section G.4).

**Supervised Contrastive Learning Methods.** The accuracy from three classifier learning strategies on SCL representations is presented in Table 2: (i) standard linear probing with learnable weights and bias, (ii) normalized linear probing using NormFace loss, and (iii) fixed prototypes computed as class-mean embeddings. Fixed prototypes match linear probing performance on 3 of 4 datasets, and mark a considerable +2.0% improvement on ImageNet-100 **requiring only $N$ forward passes versus $T \times N$ for training-based methods, where $T$ is the number of epochs. Normalized lin-

ear probing achieves comparable accuracy to standard linear probing, validating that the *discriminative information in SCL features resides primarily in their angular structure* rather than magnitude or biases. These findings validate that angular structure alone suffices for discrimination in well-trained representations, enabling *training-free classification* in SCL via fixed prototypes that **eliminate huge computational costs** by discarding a, typically hours long, training phase.

## 5.2. Quantifying Neural Collapse

We quantify NC1–NC3 with complementary, condition-specific metrics; we omit NC4 (bias collapse) as our models enforce zero bias by design.

**Effective Rank (NC1, NC2).** For matrix $\mathbf{A}$ with singular values $\{\sigma_i\}$ the effective rank (Roy & Vetterli, 2007) is defined as $\text{erank}(\mathbf{A}) = \exp\{-\sum_i p_i \log p_i\}$ where $p_i = \sigma_i / \sum_j \sigma_j$. We compute the intra and inter class effective ranks (Zhang et al., 2024) as: $\text{erank}_{\text{intra}} = \frac{1}{K}\sum_{c=1}^{K} \text{erank}(\text{Cov}[\mathbf{z}_i - \mu_c \mid y_i = c])$ and $\text{erank}_{\text{inter}} = \text{erank}(\text{Cov}[\mu_c - \mu_G])$, where Cov is the covariance matrix. These metrics quantify **NC1** (within-class variability collapse): $\text{erank}_{\text{intra}} \to 0$ indicates $\mathbf{z}_i \to \mu_{y_i}$, and **NC2** (ETF structure): Zhang et al. (2024) proved that when $\text{erank}_{\text{inter}} = K - 1$ the class means form a simplex with equal pairwise angles. We also report $\text{erank}(\mathbf{W})$ to assess whether classifier weights approximate an equiangular tight frame (ETF).

**Alignment (NC3).** We quantify feature–classifier alignment by $\frac{1}{N}\sum_{i=1}^{N} \|\mathbf{z}_i - \mathbf{w}_{y_i}\|_2^2$ and also report instance-to-instance alignment to probe per-class collapse.

**Information Metrics (NC2, NC3).** For normalized Gram matrices $\mathbf{G}_W$ (weights), $\mathbf{G}_M$ (class means) and H being the matrix entropy, Song et al. (2024) connects Neural Collapse to the metrics:

$$\text{MIR} = \frac{\text{H}(\mathbf{G}_W) + \text{H}(\mathbf{G}_M) - \text{H}(\mathbf{G}_W \odot \mathbf{G}_M)}{\min\{\text{H}(\mathbf{G}_W), \text{H}(\mathbf{G}_M)\}},$$

$$\text{HDR} = \frac{|\text{H}(\mathbf{G}_W) - \text{H}(\mathbf{G}_M)|}{\max\{\text{H}(\mathbf{G}_W), \text{H}(\mathbf{G}_M)\}}.$$

These capture the information-theoretic signatures of **NC2** and **NC3** where under full collapse MIR $\to$ 1 and HDR $\to$ 0, reflecting perfect structural alignment.

In Table 3 four key findings are revealed: **(i) CE fails to achieve NC:** high intra-class variance (erank 22.5/96.4), suboptimal inter-class separation (erank 8.6/57.1 vs. theoretical K-1=9/99), and poor weight-feature alignment (w-inst 0.59/0.83, inst-inst 0.69/1.05). **(ii) Normalized softmax losses satisfy NC2-NC3** since they achieve perfect inter-class separation (erank 9.0/99.0), near-zero alignment

*Table 3.* **NC metrics** on CIFAR-10/100 (training). **Bold** marks the best within each learning family; **green** marks the overall best per dataset. Theoretical optima: Intra ER 0/0, Inter ER 9/99, Weights ER 9/99, Weight Align 0/0, Instance Align 0/0, MIR 1/1, HDR 0/0.

| Learning Family | Method | Effective Rank | | | Alignment | | Information Theory Metrics | |
|---|---|---|---|---|---|---|---|---|
| | | Intra ↓ | Inter ↑ | Weights ↑ | Weight ↓ | Instance ↓ | MIR ↑ | HDR ↓ |
| | CE | 22.5 / 96.4 | 8.6 / 57.1 | 8.9 / 89.7 | 0.59 / 0.83 | 0.69 / 1.05 | **0.98** / 0.97 | 0.03 / 0.13 |
| | ETF + DR | 9.00 / 18.4 | 8.90 / 94.8 | **9.00 / 99.00** | 0.58 / 0.59 | 0.59 / 0.61 | **0.98** / **1.00** | **0.02** / **0.11** |
| CLASSIFIER | NormFace | 10.5 / 13.6 | **9.0** / 96.2 | **9.0** / 96.1 | 0.12 / **0.01** | 0.14 / 0.06 | 0.95 / **1.00** | 0.04 / 0.30 |
| LEARNING | NTCE | 9.0 / 12.6 | **9.0** / **99.0** | 8.9 / 98.9 | **0.08** / **0.01** | **0.10** / **0.05** | 0.96 / **1.00** | 0.05 / 0.30 |
| | NONL | **4.0** / **11.4** | **9.0** / **99.0** | **9.0** / **99.0** | 0.11 / **0.01** | 0.16 / 0.06 | 0.95 / **1.00** | 0.05 / 0.30 |
| CONTRASTIVE | SCL (w probing) | **4.5** / **7.5** | **9.0** / **66.7** | 8.3 / **77.8** | 0.99 / 1.03 | **0.10** / **0.34** | 0.99 / **0.95** | **0.07** / **0.11** |
| LEARNING | SCL (w/o probing) | **4.5** / **7.5** | **9.0** / **66.7** | **9.0** / 66.5 | **0.00** / **0.00** | **0.10** / **0.34** | **1.00** / 0.87 | 0.09 / 0.14 |

*Table 4.* **Transfer Learning Results**. Numbers are top-1 accuracy (%) for all datasets except VOC2007 (mAP). Best per column in **green**.

| Method | Food | CIFAR10 | CIFAR100 | Cars | DTD | Pets | Flowers | VOC2007 | Mean |
|---|---|---|---|---|---|---|---|---|---|
| CE | 68.0 | 88.6 | 67.7 | 25.9 | 69.9 | 67.0 | 81.4 | 67.1 | **67.0** |
| ETF + DR | 57.2 | 83.9 | 54.3 | 9.8 | 64.0 | 49.2 | 58.3 | 60.0 | **54.6** |
| NormFace | 69.8 | 89.8 | 69.7 | 29.7 | **70.1** | 69.7 | 83.2 | 67.9 | **68.7** |
| NTCE | 69.3 | 89.9 | 69.8 | 28.1 | 70.0 | 69.4 | 81.9 | 67.8 | **68.3** |
| NONL | **70.7** | **90.0** | **71.0** | **38.1** | 69.4 | **72.9** | **85.2** | **68.3** | **70.7** |
| Δ(NONL−CE) | +4.0% | +1.6% | +4.9% | +47.1% | −0.7% | +8.8% | +4.7% | +1.8% | **+5.5%** |

errors (NTCE: w-inst 0.08/0.01, inst-inst 0.10/0.05), and optimal weight dimensionality matching the simplex ETF, with **NONL being the overall best** mostly due to its better intra class structure. **(iii) SCL with linear probing violates NC3:** despite superior within-class collapse (erank 4.5/7.5), inter-class structure degrades (erank 9.0/66.7) and classifier-feature alignment fails (w-inst 0.99/1.03). **(iv) Fixed prototypes restore NC3 in SCL:** removing the trainable classifier enforces perfect alignment by construction, though inter-class separation remains suboptimal. In appendix Table 7 we further show that **(v)** our objectives match cross-entropy's NC metrics with **substantially fewer training iterations**.

While CE and ETF+ DR attain slightly better MIR/HDR values than our normalized losses, these information-theoretic metrics primarily reflect the overall entropy/redundancy of the representation, not the NC geometry itself. In our case, CE appears to preserve a bit more raw variability, but organizes it in a less NC-like, less prototype-structured way (higher intra-class effective rank, weaker alignment), whereas our normalized losses reshape the same information into a cleaner NC geometry. As our downstream experiments show in Section 5.3, this structured organization is more beneficial for transfer, long-tailed performance, and robustness, even though CE may capture slightly more "information" by these metrics.

### 5.3. Practical Benefits of Collapsed Representations

**Transfer learning.** We first ask whether representations that lie closer to the NC regime are more generalizable to unseen tasks. Following typical pipelines (Khosla et al., 2020), we freeze the pretrained encoder for each loss and train a linear classifier (or detection head for VOC07) on eight diverse downstream datasets. As shown in Table 4, *NONL consistently yields strong transfer performance*: it attains the best accuracy on 7/8 datasets and delivers a +5.5% relative improvement in mean performance over CE, while NTCE also consistently exceeds CE. These results confirm prior works (Bartlett et al., 2017; Galanti et al., 2021; Neyshabur et al., 2018; Papyan et al., 2020) that explicitly encouraging NC-like geometry produces features that generalize better beyond the pretraining distribution.

**Long-tailed classification.** We next examine robustness to class imbalance using standard evaluation pipelines (Yang et al., 2022). For CIFAR-10-LT and CIFAR-100-LT, we construct long-tailed versions with three imbalance ratios and train all models directly on the imbalanced data. Table 6 shows that our NC-inducing objectives substantially improve minority-class performance: on CIFAR-100-LT, NONL outperforms CE by +3.4%, +7.7%, and +7.0% under increasing imbalance, with gains up to +8.7% across CIFAR-10/100-LT, and also surpasses the ETF+DR baseline (Yang et al., 2022). This validates the literature (Yang et al., 2022) that enforcing NC-like geometric structure helps maintain

*Table 5.* **Robustness Results.** Clean error, mCE, and corruption error (%) on ImageNet-C. Best per column in **green**.

| Network | Error | mCE | Noise | | | Blur | | | | Weather | | | | Digital | | | |
|---|---|---|---|---|---|---|---|---|---|---|---|---|---|---|---|---|---|
| | | | Gauss | Shot | Impulse | Defoc | Glass | Motion | Zoom | Snow | Frost | Fog | Bright | Contrast | Elastic | Pixel | JPEG |
| CE | 25.0 | 80.1 | 75 | 78 | 83 | **84** | 94 | 86 | 88 | 80 | 78 | 68 | 62 | 66 | 96 | 84 | 80 |
| ETF + DR | 24.6 | 79.2 | 75 | 78 | 82 | 86 | 94 | 85 | 86 | **76** | 77 | 66 | 62 | 67 | 93 | 80 | 82 |
| NormFace | 23.6 | 77.8 | 74 | 76 | 82 | **84** | **93** | **81** | **85** | **76** | **75** | **64** | **60** | **65** | **91** | 81 | 80 |
| NTCE | **23.3** | **77.6** | 73 | 76 | **80** | **84** | **93** | 83 | **85** | **76** | **75** | 65 | **60** | **65** | 93 | **78** | **79** |
| NONL | 23.5 | 77.8 | **72** | **75** | **80** | 85 | **93** | 83 | **85** | **76** | **75** | **64** | **60** | 66 | **91** | 81 | 81 |

*Table 6.* **Class Imbalance.** Performance under on CIFAR-10-LT/100-LT vs. imbalance ratio $\tau$ (Yang et al., 2022). Best per column in **green**.

| | CIFAR-10-LT | | | CIFAR-100-LT | | |
|---|---|---|---|---|---|---|
| Method | $\tau = 0.1$ | 0.02 | 0.01 | $\tau = 0.1$ | 0.02 | 0.01 |
| CE | 88.1 | 76.8 | 70.2 | 55.5 | 41.5 | 37.4 |
| ETF + DR | 88.0 | 77.8 | 71.3 | 54.4 | 40.6 | 36.2 |
| NormFace | 88.0 | 79.0 | 74.3 | 54.6 | 40.1 | 35.9 |
| NTCE | 88.8 | **80.8** | **77.3** | 56.8 | 43.7 | 39.0 |
| NONL | **89.2** | 80.5 | 76.3 | **57.4** | **44.7** | **40.0** |
| $\Delta(\text{NONL} - \text{CE})$ | +1.2% | +4.8% | +8.7% | +3.4% | +7.7% | +7.0% |

class separability even when minority classes are severely underrepresented, complementing the improvements observed in transfer.

**Out-of-distribution robustness.** Finally, we evaluate robustness to common corruptions on ImageNet-C (Hendrycks & Dietterich, 2019). Models are trained on clean ImageNet-1K only and evaluated on corrupted variants, reporting clean top-1 error and mean Corruption Error (mCE) normalized as in Hendrycks & Dietterich (2019). As summarized in Table 5, our normalized losses reduce mCE compared to CE while also improving clean accuracy. Thus, NC-inducing objectives not only improve in-distribution performance, but as reported in the literature (Ding et al., 2020; Fawzi et al., 2016; Hein & Andriushchenko, 2017), also yield representations that are more robust to distribution shift, in line with their benefits for transfer and long-tailed recognition.

**5.4. Limitations**

Our framework draws several limitations from the related literature. **(i)** Like other contrastive methods, NTCE and NONL benefit from larger batches to provide sufficient in-batch negatives (Section G.4); we explicitly acknowledge that needing larger batches is a practical drawback, but this is a typical limitation of contrastive methods (including SCL (Khosla et al., 2020) and many SSL methods (Chen et al., 2020)). Recent SSL methods (memory banks, queues, momentum encoders (He et al., 2020)) suggest promising ways to increase the number of negatives without linearly scaling batch size, which we view as a natural direction for future extensions of NONL. **(ii)** Our theory

rests on balanced-class UFM/LPM assumptions; while recent work formally justifies UFM as a faithful proxy for deep ResNet/Transformer training (Súkeník et al., 2024; 2025), end-to-end training dynamics are not directly characterized. **(iii)** For SCL specifically, Kini et al. (2024) show that when using ReLU activations the optimum is an orthogonal frame with collapsed in-class representations rather than a centered simplex ETF; the minimizer-set equivalence in Theorem 4.2 still holds with this alternative minimizer, but the resulting geometry differs from the standard NC simplex. **(iv)** In the standard transfer setting we evaluate, knowledge transfers from large-scale to smaller downstream tasks (Chen et al., 2020; Khosla et al., 2020); the recent line on coarse-to-fine transfer (Chen et al., 2022; Kornblith et al., 2021) argues that collapsed representations may struggle to distinguish fine-grained sub-classes within a coarse class.

# 6. Conclusion

In this work, we address the mismatch between the theoretical optima of supervised objectives and their behavior in practice. Constraining learning to the unit hypersphere removes the radial degeneracy of cross-entropy and unifies normalized softmax and supervised contrastive learning (SCL) as a single prototype-contrast paradigm. Building on this view, we propose two objectives (NTCE and NONL) that accelerate convergence to Neural Collapse. Theoretically, we prove SCL already yields an optimal prototype classifier during contrastive training, eliminating the typical linear probing phase. Empirically, across four benchmarks including ImageNet-1K, our methods surpass CE accuracy, reach ≥95% on NC metrics, and translate to practical benefits with better capabilities on transfer learning, long-tailed classification and robustness. Overall, supervised learning is recast as prototype-based classification on the hypersphere, narrowing the theory–practice gap while simplifying and speeding up training.

# Acknowledgements

Panagiotis Koromilas was supported by the Hellenic Foundation for Research and Innovation (HFRI) under the

4th Call for HFRI PhD Fellowships (Fellowship Number: 10816). Yannis Panagakis was supported by the project MIS 5154714 of the National Recovery and Resilience Plan Greece 2.0 funded by the European Union under the NextGenerationEU Program. Mihalis Nicolaou was supported by the TensorICE project (EXCELLENCE/0524/0407), implemented under the social cohesion programme "THALIA 2021-2027", co-funded by the European Union through the Research and Innovation Foundation. This work was also supported by computing time awarded on the Cyclone supercomputer of the High Performance Computing Facility of The Cyprus Institute.

## Impact Statement

This paper presents work whose goal is to advance the field of machine learning. Specifically, our work improves the understanding of the inner workings of supervised representation learning and proposes methods to improve several aspects of performance and optimization. There are many potential societal consequences of our work, none of which we feel must be specifically highlighted here.

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

## A. Neural Collapse Optimality of Normalized Objectives

We adopt the balanced unconstrained-features / layer-peeled model (UFM/LPM) (Tirer & Bruna, 2022; Yaras et al., 2022). The last-layer features $z_i \in \mathbb{R}^d$ and classifier weights $w_c \in \mathbb{R}^d$ are free optimization variables. We work with their $\ell_2$-normalized versions

$$u_i = \frac{z_i}{\|z_i\|}, \qquad \hat{w}_c = \frac{w_c}{\|w_c\|},$$

so that $\|u_i\| = \|\hat{w}_c\| = 1$ and

$$S_{ic} := u_i^\top \hat{w}_c.$$

There are $K$ classes and $M$ training samples, and the dataset is balanced: each class $c$ has index set

$$I_c := \{i : y_i = c\} \quad \text{with} \quad |I_c| = n = M/K.$$

We assume $d \geq K$.

**Normalized CE–based losses.** We consider three normalized cross-entropy–based losses with temperature $\tau > 0$:

$$\mathcal{L}_{\text{NF}} = -\frac{1}{M} \sum_{i=1}^M \log \frac{\exp(S_{i,y_i}/\tau)}{\sum_{c=1}^K \exp(S_{ic}/\tau)}, \tag{7}$$

$$\mathcal{L}_{\text{NTCE}} = -\frac{1}{M} \sum_{i=1}^M \log \frac{\exp(S_{i,y_i}/\tau)}{\sum_{j=1}^M \exp(S_{j,y_i}/\tau)}, \tag{8}$$

$$\mathcal{L}_{\text{NONL}} = -\frac{1}{M} \sum_{i=1}^M \log \frac{\exp(S_{i,y_i}/\tau)}{\sum_{j:\, y_j \neq y_i} \exp(S_{j,y_i}/\tau)}. \tag{9}$$

**Neural Collapse properties.** We say a configuration exhibits *Neural Collapse* (NC) if there exist unit vectors $\mu_1, \ldots, \mu_K \in \mathbb{R}^d$ such that:

(NC1) (Within-class collapse) $u_i = \mu_{y_i}$ for all $i$.

(NC2) (Simplex ETF) the vectors $\{\mu_c\}$ form a centered regular simplex in a $(K-1)$–dimensional subspace:

$$\|\mu_c\| = 1 \quad \text{and} \quad \mu_c^\top \mu_{c'} = -\frac{1}{K-1} \quad \forall c \neq c'.$$

(NC3) (Classifier–mean alignment) $\hat{w}_c = \mu_c$ for all $c$.

At such a configuration the (normalized-feature) class means $\hat{\mu}_c := \frac{1}{n} \sum_{i \in I_c} u_i$ coincide with $\mu_c$ and are therefore unit norm.

**Theorem A.1** (NC optimality of normalized CE–based losses). *In the balanced UFM/LPM setting above with $d \geq K$, every global minimizer of $\mathcal{L}_{\text{NF}}$, $\mathcal{L}_{\text{NTCE}}$, and $\mathcal{L}_{\text{NONL}}$ satisfies NC1–NC3, up to a global rotation and permutation of class labels.*

We now analyze the three losses in turn.

**NormFace**

Yaras et al. (2022) study the constrained UFM problem

$$\min_{H,W} \frac{1}{M} \sum_{i=1}^M \text{CE}\big(\tau' W^\top h_i, y_i\big) \quad \text{s.t.} \ \|h_i\| = 1, \ \|w_c\| = 1, \tag{10}$$

with $\tau' > 0$, where CE is the standard cross-entropy.

**Lemma A.2** (NormFace $\equiv$ Yaras et al.). *Set $\tau' = 1/\tau$, and identify $h_i = u_i$ and $w_c = \hat{w}_c$. Then $\mathcal{L}_{\text{NF}}$ coincides with equation 10, and $\arg\min \mathcal{L}_{\text{NF}}$ equals the set of global minimizers of equation 10 over unit-norm features and weights.*

*Proof.* With $\boldsymbol{h}_i = \boldsymbol{u}_i$, $\boldsymbol{w}_c = \hat{\boldsymbol{w}}_c$ and $\tau' = 1/\tau$, we have $\left(\tau' W^\top \boldsymbol{h}_i\right)_c = S_{ic}/\tau$, so the summand in equation 10 is exactly $-\log \frac{\exp(S_{i,y_i}/\tau)}{\sum_c \exp(S_{ic}/\tau)}$, which is the $i$th summand in $\mathcal{L}_{\mathrm{NF}}$. Averaging over $i$ gives the claim. $\qquad\square$

Theorem 3.1 of Yaras et al. (2022) states that, under balanced labels and $d \geq K$, every global minimizer of equation 10 satisfies NC1–NC3. Together with Lemma A.2, this implies that every global minimizer of $\mathcal{L}_{\mathrm{NF}}$ satisfies NC1–NC3.

**NTCE**

We now show that every global minimizer of $\mathcal{L}_{\mathrm{NTCE}}$ satisfies NC1–NC3. The proof follows the same three-step pattern we later use for NONL: we first reduce to a class-level objective depending only on class means and weights, then view this function as a contrastive loss of La/Lc type from (Koromilas et al., 2024) and apply its respective minimizer characterization of at the class level, and finally lift the resulting structure back to the sample level.

Recall that

$$\mathcal{L}_{\mathrm{NTCE}} = -\frac{1}{M} \sum_{i=1}^{M} \log \frac{\exp(\boldsymbol{u}_i^\top \hat{\boldsymbol{w}}_{y_i}/\tau)}{\sum_{j=1}^{M} \exp(\boldsymbol{u}_j^\top \hat{\boldsymbol{w}}_{y_i}/\tau)} = \frac{1}{M} \sum_{i=1}^{M} \ell_i^{\mathrm{NTCE}},$$

with per-sample loss

$$\ell_i^{\mathrm{NTCE}} := -\log \frac{\exp(\boldsymbol{u}_i^\top \hat{\boldsymbol{w}}_{y_i}/\tau)}{\sum_{j=1}^{M} \exp(\boldsymbol{u}_j^\top \hat{\boldsymbol{w}}_{y_i}/\tau)}.$$

We again work in the balanced setting $|I_c| = n = M/K$ and $\|\boldsymbol{u}_i\| = \|\hat{\boldsymbol{w}}_c\| = 1$.

**Step 1: reduction to class means.**

**Lemma A.3** (NTCE reduction via class means). *Assume balanced labels, $|I_c| = n = M/K$ for all $c$. For any configuration $\{\boldsymbol{u}_i\}, \{\hat{\boldsymbol{w}}_c\}$ with $\|\boldsymbol{u}_i\| = \|\hat{\boldsymbol{w}}_c\| = 1$ define the normalized-feature class means*

$$\hat{\boldsymbol{\mu}}_c := \frac{1}{n} \sum_{j \in I_c} \boldsymbol{u}_j.$$

*Then*

$$\mathcal{L}_{\mathrm{NTCE}}(\{\boldsymbol{u}_i\}, \{\hat{\boldsymbol{w}}_c\}) \geq L_{\mathrm{NTCE}}^{\mathrm{cls}}(\{\hat{\boldsymbol{\mu}}_c\}, \{\hat{\boldsymbol{w}}_c\}),$$

*where the class-level loss is*

$$L_{\mathrm{NTCE}}^{\mathrm{cls}} := -\frac{1}{K\tau} \sum_{c=1}^{K} \hat{\boldsymbol{w}}_c^\top \hat{\boldsymbol{\mu}}_c + \frac{1}{K} \sum_{c=1}^{K} \log \left( \sum_{c'=1}^{K} n \, \exp\left(\hat{\boldsymbol{w}}_c^\top \hat{\boldsymbol{\mu}}_{c'}/\tau\right) \right). \tag{11}$$

*Moreover, the inequality is tight if and only if, for every ordered pair $(c, c')$, the logits $\hat{\boldsymbol{w}}_c^\top \boldsymbol{u}_j$ are constant over $j \in I_{c'}$, i.e.*

$$\hat{\boldsymbol{w}}_c^\top \boldsymbol{u}_j = \hat{\boldsymbol{w}}_c^\top \hat{\boldsymbol{\mu}}_{c'} \quad \text{for all } j \in I_{c'}.$$

*Proof.* Fix a configuration $\{\boldsymbol{u}_i\}, \{\hat{\boldsymbol{w}}_c\}$ and define the class means $\hat{\boldsymbol{\mu}}_c$ as above. Using the balanced labels, write $\mathcal{L}_{\mathrm{NTCE}}$ as an average over classes. For $i \in I_c$ we have

$$\ell_i^{\mathrm{NTCE}} = -\frac{1}{\tau} \hat{\boldsymbol{w}}_c^\top \boldsymbol{u}_i + \log \left( \sum_{j=1}^{M} \exp(\hat{\boldsymbol{w}}_c^\top \boldsymbol{u}_j/\tau) \right),$$

so

$$\mathcal{L}_{\mathrm{NTCE}} = \frac{1}{M} \sum_{c=1}^{K} \sum_{i \in I_c} \ell_i^{\mathrm{NTCE}}$$

$$= -\frac{1}{M\tau} \sum_{c=1}^{K} \sum_{i \in I_c} \hat{\boldsymbol{w}}_c^\top \boldsymbol{u}_i + \frac{1}{M} \sum_{c=1}^{K} \sum_{i \in I_c} \log \left( \sum_{j=1}^{M} \exp(\hat{\boldsymbol{w}}_c^\top \boldsymbol{u}_j/\tau) \right).$$

The denominator term inside the logarithm depends only on the anchor class $c$, not on $i$, so $\sum_{i \in I_c}$ introduces a factor of $|I_c| = n$. Using $M = nK$ and the definition of $\hat{\boldsymbol{\mu}}_c$ we obtain

$$
\mathcal{L}_{\text{NTCE}} = -\frac{1}{K\tau} \sum_{c=1}^{K} \hat{\boldsymbol{w}}_c^{\top} \Big( \frac{1}{n} \sum_{i \in I_c} \boldsymbol{u}_i \Big) + \frac{1}{K} \sum_{c=1}^{K} \log \Big( \sum_{j=1}^{M} \exp(\hat{\boldsymbol{w}}_c^{\top} \boldsymbol{u}_j / \tau) \Big)
$$

$$
= -\frac{1}{K\tau} \sum_{c=1}^{K} \hat{\boldsymbol{w}}_c^{\top} \hat{\boldsymbol{\mu}}_c + \frac{1}{K} \sum_{c=1}^{K} \log \Big( \sum_{j=1}^{M} \exp(\hat{\boldsymbol{w}}_c^{\top} \boldsymbol{u}_j / \tau) \Big).
$$

For each fixed anchor class $c$, split the denominator over classes:

$$
\sum_{j=1}^{M} \exp(\hat{\boldsymbol{w}}_c^{\top} \boldsymbol{u}_j / \tau) = \sum_{c'=1}^{K} \sum_{j \in I_{c'}} \exp(\hat{\boldsymbol{w}}_c^{\top} \boldsymbol{u}_j / \tau).
$$

For fixed $(c, c')$, the function $f_c(\boldsymbol{x}) := \exp(\hat{\boldsymbol{w}}_c^{\top} \boldsymbol{x} / \tau)$ is convex in $\boldsymbol{x}$, so by Jensen's inequality over $j \in I_{c'}$,

$$
\frac{1}{n} \sum_{j \in I_{c'}} \exp(\hat{\boldsymbol{w}}_c^{\top} \boldsymbol{u}_j / \tau) = \frac{1}{n} \sum_{j \in I_{c'}} f_c(\boldsymbol{u}_j) \ \geq \ f_c\Big( \frac{1}{n} \sum_{j \in I_{c'}} \boldsymbol{u}_j \Big) = \exp(\hat{\boldsymbol{w}}_c^{\top} \hat{\boldsymbol{\mu}}_{c'} / \tau).
$$

Multiplying by $n$ and summing over $c'$ yields

$$
\sum_{j=1}^{M} \exp(\hat{\boldsymbol{w}}_c^{\top} \boldsymbol{u}_j / \tau) \ \geq \ \sum_{c'=1}^{K} n \, \exp(\hat{\boldsymbol{w}}_c^{\top} \hat{\boldsymbol{\mu}}_{c'} / \tau).
$$

Taking logs and averaging over $c$ gives

$$
\mathcal{L}_{\text{NTCE}} \ \geq \ -\frac{1}{K\tau} \sum_{c=1}^{K} \hat{\boldsymbol{w}}_c^{\top} \hat{\boldsymbol{\mu}}_c + \frac{1}{K} \sum_{c=1}^{K} \log \left( \sum_{c'=1}^{K} n \, \exp(\hat{\boldsymbol{w}}_c^{\top} \hat{\boldsymbol{\mu}}_{c'} / \tau) \right) = L_{\text{NTCE}}^{\text{cls}}.
$$

Jensen's inequality is tight for a given pair $(c, c')$ if and only if the arguments of $f_c$ are constant over $j \in I_{c'}$, i.e. if and only if $\hat{\boldsymbol{w}}_c^{\top} \boldsymbol{u}_j$ is constant in $j$ on $I_{c'}$. In that case this constant must equal $\hat{\boldsymbol{w}}_c^{\top} \hat{\boldsymbol{\mu}}_{c'}$. Tightness for all $c, c'$ gives the stated condition. $\qquad \square$

Thus, for any configuration of unit features and weights, the NTCE loss is lower-bounded by the class-level objective $L_{\text{NTCE}}^{\text{cls}}$ depending only on the $K$ class means $\hat{\boldsymbol{\mu}}_c$ and the $K$ classifier weights $\hat{\boldsymbol{w}}_c$, and Lemma A.3 precisely characterizes when this lower bound is tight (blockwise constant logits).

It is convenient to separate out the constant $\log n$ factor, and to view the class means and weights abstractly as unit vectors. Define the *normalized* class-level NTCE loss

$$
\tilde{L}_{\text{NTCE}}^{\text{cls}}(\{\hat{\boldsymbol{\mu}}_c\}, \{\hat{\boldsymbol{w}}_c\}) := -\frac{1}{K\tau} \sum_{c=1}^{K} \hat{\boldsymbol{w}}_c^{\top} \hat{\boldsymbol{\mu}}_c + \frac{1}{K} \sum_{c=1}^{K} \log \left( \sum_{c'=1}^{K} \exp \big( \hat{\boldsymbol{w}}_c^{\top} \hat{\boldsymbol{\mu}}_{c'} / \tau \big) \right),
$$

so that

$$
L_{\text{NTCE}}^{\text{cls}} = \log n + \tilde{L}_{\text{NTCE}}^{\text{cls}}.
$$

In what follows, we treat the pairs $(\hat{\boldsymbol{\mu}}_c, \hat{\boldsymbol{w}}_c)$ as free variables on the unit sphere and, to lighten notation, write $\boldsymbol{\mu}_c := \hat{\boldsymbol{\mu}}_c$ and $\boldsymbol{w}_c := \hat{\boldsymbol{w}}_c$.

**Step 2: analysis of the class-level problem.** For each class $c$ we view the $c$th summand in $\tilde{L}^{\text{cls}}_{\text{NTCE}}$ as a standard contrastive loss of La/Lc type (Koromilas et al., 2024), with

$$q_c = \boldsymbol{w}_c \quad \text{(anchor)}, \qquad k_c^+ = \boldsymbol{\mu}_c \quad \text{(positive)}, \qquad \{k_c^- = \boldsymbol{\mu}_{c'} : c' \neq c\} \quad \text{(negatives)}.$$

The per-class alignment and contrastive terms are

$$L_{\text{a}}(q_c, k_c^+) = -\frac{1}{\tau} q_c^\top k_c^+, \qquad L_{\text{c}}(q_c, \{k_c^-\}) = \log\Big(\sum_{c'=1}^{K} \exp(q_c^\top k_{c'}^- / \tau)\Big).$$

The La/Lc framework requires $L_{\text{a}}$ to be strictly decreasing in similarity and $L_{\text{c}}$ to be convex and strictly increasing in similarity. These conditions hold here:

- $q_c^\top k_c^+$ enters $L_{\text{a}}$ linearly with a negative coefficient, so $L_{\text{a}}$ decreases as $q_c^\top k_c^+$ increases.

- $L_{\text{c}}$ is a log-sum-exp of the similarities $q_c^\top k_c^- / \tau$, hence convex and strictly increasing in each similarity argument.

Therefore we may invoke the minimizer characterization for La/Lc losses. By Theorem 4.1 and Appendix B.1 of Koromilas et al. (2024), provided $d \geq K$, the global minimizers of $\tilde{L}^{\text{cls}}_{\text{NTCE}}$ over unit vectors satisfy:

- **Perfect alignment:** $\boldsymbol{\mu}_c = \boldsymbol{w}_c$ for all $c$.

- **Simplex ETF structure:** the directions $\{\boldsymbol{\mu}_c\}_{c=1}^K$ form a centered regular simplex equiangular tight frame in a $(K-1)$–dimensional subspace:
$$\|\boldsymbol{\mu}_c\| = 1, \qquad \boldsymbol{\mu}_c^\top \boldsymbol{\mu}_{c'} = -\frac{1}{K-1} \quad \forall c \neq c'.$$

In particular, there exists a simplex ETF $\{\boldsymbol{\mu}_c\}_{c=1}^K \subset \mathbb{R}^d$ such that $\boldsymbol{\mu}_c = \boldsymbol{w}_c$ is a global minimizer of $\tilde{L}^{\text{cls}}_{\text{NTCE}}$, unique up to a global rotation and permutation of the class indices.

**Step 3: lifting back to the sample level.** We now relate these class-level minimizers back to the original sample-level NTCE objective and derive the NC structure of its global minimizers.

*Existence of Neural Collapse minimizers.* Let $\{\boldsymbol{\mu}_c\}_{c=1}^K$ be a simplex ETF and set

$$\hat{\boldsymbol{w}}_c := \boldsymbol{\mu}_c, \qquad \boldsymbol{u}_i := \boldsymbol{\mu}_{y_i} \quad \text{for all } i.$$

This configuration satisfies NC1–NC3 by construction: within each class $c$, all normalized features collapse to $\boldsymbol{\mu}_c$ (NC1), the vectors $\{\boldsymbol{\mu}_c\}$ form a centered simplex ETF (NC2), and $\hat{\boldsymbol{w}}_c = \boldsymbol{\mu}_c$ (NC3). In particular, the feature class means are $\hat{\boldsymbol{\mu}}_c = \boldsymbol{\mu}_c$.

Moreover, for this configuration the Jensen inequalities in Lemma A.3 are tight: for any anchor class $c$ and any class $c'$, we have $\hat{\boldsymbol{w}}_c^\top \boldsymbol{u}_j = \boldsymbol{\mu}_c^\top \boldsymbol{\mu}_{c'}$ for all $j \in I_{c'}$, so the logits are constant within each class. Hence

$$\mathcal{L}_{\text{NTCE}} = L^{\text{cls}}_{\text{NTCE}}(\{\hat{\boldsymbol{\mu}}_c\}, \{\hat{\boldsymbol{w}}_c\}) = \log n + \tilde{L}^{\text{cls}}_{\text{NTCE}}(\{\boldsymbol{\mu}_c\}, \{\boldsymbol{\mu}_c\}).$$

Since $\{\boldsymbol{\mu}_c\}, \{\boldsymbol{\mu}_c\}$ is a global minimizer of $\tilde{L}^{\text{cls}}_{\text{NTCE}}$, this shows that

$$\inf_{\{\boldsymbol{u}_i\}, \{\hat{\boldsymbol{w}}_c\}} \mathcal{L}_{\text{NTCE}} \leq \log n + \inf_{\{\boldsymbol{\mu}_c\}, \{\boldsymbol{w}_c\}} \tilde{L}^{\text{cls}}_{\text{NTCE}}.$$

*Structure of arbitrary global minimizers.* Conversely, let $(\{\boldsymbol{u}_i^\star\}, \{\hat{\boldsymbol{w}}_c^\star\})$ be any global minimizer of $\mathcal{L}_{\text{NTCE}}$, and let

$$\hat{\boldsymbol{\mu}}_c^\star := \frac{1}{n} \sum_{j \in I_c} \boldsymbol{u}_j^\star$$

be the corresponding class means. Lemma A.3 gives

$$\mathcal{L}_{\text{NTCE}}(\{\boldsymbol{u}_i^\star\}, \{\hat{\boldsymbol{w}}_c^\star\}) \geq L^{\text{cls}}_{\text{NTCE}}(\{\hat{\boldsymbol{\mu}}_c^\star\}, \{\hat{\boldsymbol{w}}_c^\star\}) = \log n + \tilde{L}^{\text{cls}}_{\text{NTCE}}(\{\hat{\boldsymbol{\mu}}_c^\star\}, \{\hat{\boldsymbol{w}}_c^\star\}).$$

On the other hand, from the ETF construction above we know that

$$\inf_{\{\boldsymbol{u}_i\},\{\hat{\boldsymbol{w}}_c\}} \mathcal{L}_{\text{NTCE}} \leq \log n + \inf_{\{\boldsymbol{\mu}_c\},\{\boldsymbol{w}_c\}} \tilde{L}_{\text{NTCE}}^{\text{cls}}.$$

Since $(\{\boldsymbol{u}_i^\star\},\{\hat{\boldsymbol{w}}_c^\star\})$ achieves the global infimum, the two displays must be equalities. Therefore:

- $\tilde{L}_{\text{NTCE}}^{\text{cls}}(\{\hat{\boldsymbol{\mu}}_c^\star\},\{\hat{\boldsymbol{w}}_c^\star\})$ attains the global minimum of $\tilde{L}_{\text{NTCE}}^{\text{cls}}$, so by the La/Lc minimizer characterization we must have, up to a global rotation and permutation of class labels,

$$\hat{\boldsymbol{\mu}}_c^\star = \hat{\boldsymbol{w}}_c^\star \quad \text{for all } c, \qquad \{\hat{\boldsymbol{\mu}}_c^\star\} \text{ form a centered simplex ETF.}$$

- Lemma A.3 must be tight at the minimizer, so the Jensen equalities hold for all $(c, c')$: for every anchor class $c$ and every class $c'$, the logits $\hat{\boldsymbol{w}}_c^{\star\top} \boldsymbol{u}_j^\star$ are constant over $j \in I_{c'}$, equal to $\hat{\boldsymbol{w}}_c^{\star\top} \hat{\boldsymbol{\mu}}_{c'}^\star$.

Let $S := \text{span}\{\hat{\boldsymbol{w}}_1^\star, \ldots, \hat{\boldsymbol{w}}_K^\star\}$, which is the $(K-1)$–dimensional simplex-ETF subspace. Fix a class $c'$ and $j \in I_{c'}$. For every $c \neq c'$, tightness of Jensen gives

$$\hat{\boldsymbol{w}}_c^{\star\top}\left(\boldsymbol{u}_j^\star - \hat{\boldsymbol{\mu}}_{c'}^\star\right) = 0.$$

Since $\{\hat{\boldsymbol{w}}_c^\star\}_{c=1}^K$ form a centered simplex ETF in the $(K-1)$–dimensional subspace $S$ and satisfy $\sum_{c=1}^K \hat{\boldsymbol{w}}_c^\star = 0$, any $K-1$ of them are linearly independent and thus span $S$. In particular, the set $\{\hat{\boldsymbol{w}}_c^\star : c \neq c'\}$ spans $S$, so $\boldsymbol{u}_j^\star - \hat{\boldsymbol{\mu}}_{c'}^\star$ is orthogonal to $S$, and hence the orthogonal projection of $\boldsymbol{u}_j^\star$ onto $S$ equals $\hat{\boldsymbol{\mu}}_{c'}^\star$.

But both $\boldsymbol{u}_j^\star$ and $\hat{\boldsymbol{\mu}}_{c'}^\star = \hat{\boldsymbol{w}}_{c'}^\star$ are unit vectors, and $\hat{\boldsymbol{\mu}}_{c'}^\star \in S$. The only way for a unit vector to have a unit-norm projection onto $S$ is to lie in $S$ itself and coincide with its projection, so we must have

$$\boldsymbol{u}_j^\star = \hat{\boldsymbol{\mu}}_{c'}^\star \quad \text{for all } j \in I_{c'}.$$

Thus within each class all features collapse to a single unit direction (NC1), these $K$ directions form a centered simplex ETF (NC2), and the classifier weights align with the class means (NC3). Therefore every global minimizer of $\mathcal{L}_{\text{NTCE}}$ exhibits Neural Collapse, up to a global rotation and permutation of the class labels.

**NONL**

We finally treat the NONL objective equation 9. The proof proceeds by first bounding the sample-level loss by a class-level objective depending only on class means and weights, then applying the La/Lc minimizer characterization at the class level, and finally lifting this structure back to the sample level to obtain NC1–NC3.

Recall that

$$\mathcal{L}_{\text{NONL}} = -\frac{1}{M}\sum_{i=1}^M \log \frac{\exp(S_{i,y_i}/\tau)}{\sum_{j:\, y_j \neq y_i} \exp(S_{j,y_i}/\tau)} = \frac{1}{M}\sum_{i=1}^M \ell_i^{\text{NONL}},$$

with per-sample loss

$$\ell_i^{\text{NONL}} := -\log \frac{\exp(\boldsymbol{u}_i^\top \hat{\boldsymbol{w}}_{y_i}/\tau)}{\sum_{j:\, y_j \neq y_i} \exp(\boldsymbol{u}_j^\top \hat{\boldsymbol{w}}_{y_i}/\tau)}.$$

We again work in the balanced setting $|I_c| = n = M/K$ and $\|\boldsymbol{u}_i\| = \|\hat{\boldsymbol{w}}_c\| = 1$.

**Step 1: reduction to class means.** We first show that the sample-level NONL loss admits a lower bound that depends only on the $K$ normalized-feature class means and the $K$ classifier weights.

**Lemma A.4** (NONL reduction via class means). *Assume balanced labels, $|I_c| = n = M/K$ for all $c$. For any configuration $\{\boldsymbol{u}_i\}, \{\hat{\boldsymbol{w}}_c\}$ with $\|\boldsymbol{u}_i\| = \|\hat{\boldsymbol{w}}_c\| = 1$ define the normalized-feature class means*

$$\hat{\boldsymbol{\mu}}_c := \frac{1}{n}\sum_{j \in I_c} \boldsymbol{u}_j.$$

*Then*

$$\mathcal{L}_{\text{NONL}}(\{\boldsymbol{u}_i\}, \{\hat{\boldsymbol{w}}_c\}) \geq L_{\text{NONL}}^{\text{cls}}(\{\hat{\boldsymbol{\mu}}_c\}, \{\hat{\boldsymbol{w}}_c\}),$$

*where the class-level loss is*

$$L_{\text{NONL}}^{\text{cls}} := -\frac{1}{K\tau} \sum_{c=1}^{K} \hat{\boldsymbol{w}}_c^\top \hat{\boldsymbol{\mu}}_c + \frac{1}{K} \sum_{c=1}^{K} \log \left( \sum_{c' \neq c} n \, \exp\left( \hat{\boldsymbol{w}}_c^\top \hat{\boldsymbol{\mu}}_{c'} / \tau \right) \right). \tag{12}$$

*Moreover, the inequality is tight if and only if, for every ordered pair $(c, c')$ with $c' \neq c$, the "negative" logits $\hat{\boldsymbol{w}}_c^\top \boldsymbol{u}_j$ are constant over $j \in I_{c'}$, i.e.*

$$\hat{\boldsymbol{w}}_c^\top \boldsymbol{u}_j = \hat{\boldsymbol{w}}_c^\top \hat{\boldsymbol{\mu}}_{c'} \quad \text{for all } j \in I_{c'}, \ c' \neq c.$$

*Proof.* Fix a sample index $i$ with label $y_i = c$. Its NONL denominator is

$$D_i^{\text{neg}} := \sum_{j:\, y_j \neq c} \exp(\boldsymbol{u}_j^\top \hat{\boldsymbol{w}}_c / \tau) = \sum_{c' \neq c} \sum_{j \in I_{c'}} \exp(\boldsymbol{u}_j^\top \hat{\boldsymbol{w}}_c / \tau).$$

For each anchor class $c$ and negative class $c' \neq c$, consider the function

$$f_c(\boldsymbol{x}) := \exp(\hat{\boldsymbol{w}}_c^\top \boldsymbol{x} / \tau),$$

which is convex in $\boldsymbol{x}$. Applying Jensen's inequality over the negative-class samples $\{\boldsymbol{u}_j : j \in I_{c'}\}$ gives

$$\frac{1}{n} \sum_{j \in I_{c'}} \exp(\hat{\boldsymbol{w}}_c^\top \boldsymbol{u}_j / \tau) = \frac{1}{n} \sum_{j \in I_{c'}} f_c(\boldsymbol{u}_j) \ \geq \ f_c\left( \frac{1}{n} \sum_{j \in I_{c'}} \boldsymbol{u}_j \right) = \exp\left( \hat{\boldsymbol{w}}_c^\top \hat{\boldsymbol{\mu}}_{c'} / \tau \right).$$

Multiplying by $n$ and summing over $c' \neq c$ yields

$$D_i^{\text{neg}} = \sum_{c' \neq c} \sum_{j \in I_{c'}} \exp(\boldsymbol{u}_j^\top \hat{\boldsymbol{w}}_c / \tau) \ \geq \ \sum_{c' \neq c} n \, \exp(\hat{\boldsymbol{w}}_c^\top \hat{\boldsymbol{\mu}}_{c'} / \tau).$$

By definition of $\ell_i^{\text{NONL}}$,

$$\begin{aligned}
\ell_i^{\text{NONL}} &= -\log \frac{\exp(\boldsymbol{u}_i^\top \hat{\boldsymbol{w}}_c / \tau)}{D_i^{\text{neg}}} \\
&= -\frac{1}{\tau} \hat{\boldsymbol{w}}_c^\top \boldsymbol{u}_i + \log D_i^{\text{neg}} \\
&\geq -\frac{1}{\tau} \hat{\boldsymbol{w}}_c^\top \boldsymbol{u}_i + \log \left( \sum_{c' \neq c} n \, \exp(\hat{\boldsymbol{w}}_c^\top \hat{\boldsymbol{\mu}}_{c'} / \tau) \right) =: \tilde{\ell}_i.
\end{aligned}$$

This inequality is tight if and only if all Jensen steps above are equalities. For a fixed $(c, c')$ with $c' \neq c$, equality in Jensen requires that the arguments of $f_c$ be constant over $j \in I_{c'}$, i.e. $\hat{\boldsymbol{w}}_c^\top \boldsymbol{u}_j$ is constant on $I_{c'}$. Using the definition of $\hat{\boldsymbol{\mu}}_{c'}$, this constant must then equal $\hat{\boldsymbol{w}}_c^\top \hat{\boldsymbol{\mu}}_{c'}$, giving the stated tightness condition.

Finally, average $\tilde{\ell}_i$ over all samples. Using the balanced labels $|I_c| = n$ and the definition of $\hat{\boldsymbol{\mu}}_c$,

$$\begin{aligned}
\frac{1}{M} \sum_{i=1}^{M} \tilde{\ell}_i &= \frac{1}{M} \sum_{c=1}^{K} \sum_{i \in I_c} \left[ -\frac{1}{\tau} \hat{\boldsymbol{w}}_c^\top \boldsymbol{u}_i + \log \left( \sum_{c' \neq c} n \, \exp(\hat{\boldsymbol{w}}_c^\top \hat{\boldsymbol{\mu}}_{c'} / \tau) \right) \right] \\
&= -\frac{1}{M\tau} \sum_{c=1}^{K} \sum_{i \in I_c} \hat{\boldsymbol{w}}_c^\top \boldsymbol{u}_i + \frac{1}{M} \sum_{c=1}^{K} |I_c| \log \left( \sum_{c' \neq c} n \, \exp(\hat{\boldsymbol{w}}_c^\top \hat{\boldsymbol{\mu}}_{c'} / \tau) \right) \\
&= -\frac{1}{K\tau} \sum_{c=1}^{K} \hat{\boldsymbol{w}}_c^\top \hat{\boldsymbol{\mu}}_c + \frac{1}{K} \sum_{c=1}^{K} \log \left( \sum_{c' \neq c} n \, \exp(\hat{\boldsymbol{w}}_c^\top \hat{\boldsymbol{\mu}}_{c'} / \tau) \right) \\
&= L_{\text{NONL}}^{\text{cls}}(\{\hat{\boldsymbol{\mu}}_c\}, \{\hat{\boldsymbol{w}}_c\}).
\end{aligned}$$

Since $\mathcal{L}_{\text{NONL}}$ is the average of the $\ell_i^{\text{NONL}}$ and each $\ell_i^{\text{NONL}} \geq \tilde{\ell}_i$, we obtain $\mathcal{L}_{\text{NONL}} \geq L_{\text{NONL}}^{\text{cls}}$ with the stated equality condition. $\qquad \square$

As before, it is convenient to separate out the factor $n$ from the logarithm and to treat the class means and weights abstractly as unit vectors. Define the *normalized* class-level NONL loss

$$\tilde{L}^{\text{cls}}_{\text{NONL}}(\{\hat{\boldsymbol{\mu}}_c\}, \{\hat{\boldsymbol{w}}_c\}) := -\frac{1}{K\tau} \sum_{c=1}^{K} \hat{\boldsymbol{w}}_c^\top \hat{\boldsymbol{\mu}}_c + \frac{1}{K} \sum_{c=1}^{K} \log \left( \sum_{c' \neq c} \exp \left( \hat{\boldsymbol{w}}_c^\top \hat{\boldsymbol{\mu}}_{c'} / \tau \right) \right),$$

so that

$$L^{\text{cls}}_{\text{NONL}} = \log n + \tilde{L}^{\text{cls}}_{\text{NONL}}.$$

In what follows we again treat $(\hat{\boldsymbol{\mu}}_c, \hat{\boldsymbol{w}}_c)$ as free unit vectors and write $\boldsymbol{\mu}_c := \hat{\boldsymbol{\mu}}_c$ and $\boldsymbol{w}_c := \hat{\boldsymbol{w}}_c$.

**Step 2: analysis of the class-level problem.** For each class $c$ we can view the $c$th summand in $\tilde{L}^{\text{cls}}_{\text{NONL}}$ as a standard decoupled alignment/uniformity loss of La/Lc type (Koromilas et al., 2024), with:

$$q_c = \boldsymbol{w}_c \quad \text{(anchor)}, \qquad k_c^+ = \boldsymbol{\mu}_c \quad \text{(positive)}, \qquad \{k_c^- = \boldsymbol{\mu}_{c'} : c' \neq c\} \quad \text{(negatives)}.$$

The per-class alignment and contrastive terms are

$$L_{\text{a}}(q_c, k_c^+) = -\frac{1}{\tau} q_c^\top k_c^+, \qquad L_{\text{c}}(q_c, \{k_c^-\}) = \log \left( \sum_{c' \neq c} \exp(q_c^\top k_{c'}^- / \tau) \right).$$

As in the NTCE case, $L_{\text{a}}$ is strictly decreasing in similarity and $L_{\text{c}}$ is convex and strictly increasing in the similarities. Therefore we may again invoke the La/Lc minimizer characterization. By Theorem 4.1 and Appendix B.1 of Koromilas et al. (2024), provided $d \geq K$, the global minimizers of $\tilde{L}^{\text{cls}}_{\text{NONL}}$ over unit vectors satisfy:

- **Perfect alignment:** $\boldsymbol{\mu}_c = \boldsymbol{w}_c$ for all $c$.

- **Simplex ETF structure:** the directions $\{\boldsymbol{\mu}_c\}_{c=1}^{K}$ form a centered regular simplex equiangular tight frame in a $(K-1)$–dimensional subspace:

$$\|\boldsymbol{\mu}_c\| = 1, \qquad \boldsymbol{\mu}_c^\top \boldsymbol{\mu}_{c'} = -\frac{1}{K-1} \quad \forall c \neq c'.$$

In particular, there exists a simplex ETF $\{\boldsymbol{\mu}_c\}_{c=1}^{K} \subset \mathbb{R}^d$ such that $\boldsymbol{\mu}_c = \boldsymbol{w}_c$ is a global minimizer of $\tilde{L}^{\text{cls}}_{\text{NONL}}$, unique up to a global rotation and permutation of the class indices.

**Step 3: lifting back to the sample level.** We now relate these class-level minimizers back to the original sample-level NONL objective and derive the NC structure of its global minimizers.

*Existence of Neural Collapse minimizers.* Let $\{\boldsymbol{\mu}_c\}_{c=1}^{K}$ be a simplex ETF and set

$$\hat{\boldsymbol{w}}_c := \boldsymbol{\mu}_c, \qquad \boldsymbol{u}_i := \boldsymbol{\mu}_{y_i} \quad \text{for all } i.$$

This configuration satisfies NC1–NC3 by construction: within each class $c$, all normalized features collapse to $\boldsymbol{\mu}_c$ (NC1), the vectors $\{\boldsymbol{\mu}_c\}$ form a centered simplex ETF (NC2), and $\hat{\boldsymbol{w}}_c = \boldsymbol{\mu}_c$ (NC3). In particular, the feature class means are $\hat{\boldsymbol{\mu}}_c = \boldsymbol{\mu}_c$.

Moreover, for this configuration the Jensen inequalities in Lemma A.4 are tight for all $(c, c')$: for any anchor class $c$ and negative class $c' \neq c$ we have $\hat{\boldsymbol{w}}_c^\top \boldsymbol{u}_j = \boldsymbol{\mu}_c^\top \boldsymbol{\mu}_{c'}$ for all $j \in I_{c'}$, so the negative logits are constant within each negative class. Hence

$$\mathcal{L}_{\text{NONL}} = L^{\text{cls}}_{\text{NONL}}(\{\hat{\boldsymbol{\mu}}_c\}, \{\hat{\boldsymbol{w}}_c\}) = \log n + \tilde{L}^{\text{cls}}_{\text{NONL}}(\{\boldsymbol{\mu}_c\}, \{\boldsymbol{\mu}_c\}).$$

Since $\{\boldsymbol{\mu}_c\}, \{\boldsymbol{\mu}_c\}$ is a global minimizer of $\tilde{L}^{\text{cls}}_{\text{NONL}}$, this shows that

$$\inf_{\{\boldsymbol{u}_i\}, \{\hat{\boldsymbol{w}}_c\}} \mathcal{L}_{\text{NONL}} \leq \log n + \inf_{\{\boldsymbol{\mu}_c\}, \{\boldsymbol{w}_c\}} \tilde{L}^{\text{cls}}_{\text{NONL}}.$$

*Structure of arbitrary global minimizers.* Conversely, let $(\{\boldsymbol{u}_i^\star\}, \{\hat{\boldsymbol{w}}_c^\star\})$ be any global minimizer of $\mathcal{L}_{\text{NONL}}$, and let

$$\hat{\boldsymbol{\mu}}_c^\star := \frac{1}{n} \sum_{j \in I_c} \boldsymbol{u}_j^\star$$

be the corresponding class means. Lemma A.4 gives

$$\mathcal{L}_{\mathrm{NONL}}(\{\boldsymbol{u}_i^\star\}, \{\hat{\boldsymbol{w}}_c^\star\}) \;\geq\; L_{\mathrm{NONL}}^{\mathrm{cls}}(\{\hat{\boldsymbol{\mu}}_c^\star\}, \{\hat{\boldsymbol{w}}_c^\star\}) = \log n + \tilde{L}_{\mathrm{NONL}}^{\mathrm{cls}}(\{\hat{\boldsymbol{\mu}}_c^\star\}, \{\hat{\boldsymbol{w}}_c^\star\}).$$

On the other hand, from the ETF construction above we know that

$$\inf_{\{\boldsymbol{u}_i\}, \{\hat{\boldsymbol{w}}_c\}} \mathcal{L}_{\mathrm{NONL}} \;\leq\; \log n + \inf_{\{\boldsymbol{\mu}_c\}, \{\boldsymbol{w}_c\}} \tilde{L}_{\mathrm{NONL}}^{\mathrm{cls}}.$$

Since $(\{\boldsymbol{u}_i^\star\}, \{\hat{\boldsymbol{w}}_c^\star\})$ achieves this infimum, the two displays must be equalities. Therefore:

- $\tilde{L}_{\mathrm{NONL}}^{\mathrm{cls}}(\{\hat{\boldsymbol{\mu}}_c^\star\}, \{\hat{\boldsymbol{w}}_c^\star\})$ attains the global minimum of $\tilde{L}_{\mathrm{NONL}}^{\mathrm{cls}}$, so by the La/Lc minimizer characterization we must have, up to a global rotation and permutation of class labels,

$$\hat{\boldsymbol{\mu}}_c^\star = \hat{\boldsymbol{w}}_c^\star \quad \text{for all } c, \qquad \{\hat{\boldsymbol{\mu}}_c^\star\} \text{ form a centered simplex ETF.}$$

- Lemma A.4 must be tight at the minimizer, so the Jensen equalities hold for all $(c, c')$: for every anchor class $c$ and negative class $c' \neq c$, the logits $\hat{\boldsymbol{w}}_c^{\star\top} \boldsymbol{u}_j^\star$ are constant over $j \in I_{c'}$, equal to $\hat{\boldsymbol{w}}_c^{\star\top} \hat{\boldsymbol{\mu}}_{c'}^\star$.

Let $S := \mathrm{span}\{\hat{\boldsymbol{w}}_1^\star, \ldots, \hat{\boldsymbol{w}}_K^\star\}$, which is the $(K-1)$–dimensional simplex-ETF subspace. Fix a class $c'$ and $j \in I_{c'}$. For every $c \neq c'$, tightness of Jensen gives

$$\hat{\boldsymbol{w}}_c^{\star\top} \left( \boldsymbol{u}_j^\star - \hat{\boldsymbol{\mu}}_{c'}^\star \right) = 0.$$

As before, since $\{\hat{\boldsymbol{w}}_c^\star\}_{c=1}^K$ form a centered simplex ETF in $S$ and $\sum_{c=1}^K \hat{\boldsymbol{w}}_c^\star = 0$, any $K-1$ of them are linearly independent and thus span $S$. In particular, the set $\{\hat{\boldsymbol{w}}_c^\star : c \neq c'\}$ spans $S$, so $\boldsymbol{u}_j^\star - \hat{\boldsymbol{\mu}}_{c'}^\star$ is orthogonal to $S$, and hence the orthogonal projection of $\boldsymbol{u}_j^\star$ onto $S$ equals $\hat{\boldsymbol{\mu}}_{c'}^\star$.

But both $\boldsymbol{u}_j^\star$ and $\hat{\boldsymbol{\mu}}_{c'}^\star = \hat{\boldsymbol{w}}_{c'}^\star$ are unit vectors, and $\hat{\boldsymbol{\mu}}_{c'}^\star \in S$. As in the NTCE case, the only way for a unit vector to have a unit-norm projection onto $S$ is to lie in $S$ itself and coincide with its projection, so we must have

$$\boldsymbol{u}_j^\star = \hat{\boldsymbol{\mu}}_{c'}^\star \quad \text{for all } j \in I_{c'}.$$

Thus within each class all features collapse to a single unit direction (NC1), these $K$ directions form a centered simplex ETF (NC2), and the classifier weights align with the class means (NC3). Therefore every global minimizer of $\mathcal{L}_{\mathrm{NONL}}$ exhibits Neural Collapse, up to a global rotation and permutation of the class labels.

*Proof of Theorem A.1.* **NormFace:** Lemma A.2 together with Theorem 3.1 of Yaras et al. (2022) shows that every global minimizer of $\mathcal{L}_{\mathrm{NF}}$ satisfies NC1–NC3.

**NTCE:** Lemma A.3 bounds $\mathcal{L}_{\mathrm{NTCE}}$ by the class-level loss $L_{\mathrm{NTCE}}^{\mathrm{cls}}$, while the La/Lc minimizer characterization (Step 2) identifies the global minimizers of $\tilde{L}_{\mathrm{NTCE}}^{\mathrm{cls}}$ as simplex ETF configurations with $\boldsymbol{\mu}_c = \boldsymbol{w}_c$. Step 3 shows that any global minimizer of $\mathcal{L}_{\mathrm{NTCE}}$ must both attain this class-level minimum and satisfy the tightness conditions in Lemma A.3, which enforces NC1. Together these yield NC1–NC3 for all NTCE minimizers.

**NONL:** Lemma A.4 bounds $\mathcal{L}_{\mathrm{NONL}}$ by the class-level loss $L_{\mathrm{NONL}}^{\mathrm{cls}}$, while the La/Lc minimizer characterization (Step 2) identifies the global minimizers of $\tilde{L}_{\mathrm{NONL}}^{\mathrm{cls}}$ as simplex ETF configurations with $\boldsymbol{\mu}_c = \boldsymbol{w}_c$. Step 3 shows that any global minimizer of $\mathcal{L}_{\mathrm{NONL}}$ must both attain this class-level minimum and satisfy the tightness conditions in Lemma A.4, which again enforces NC1. Together these yield NC1–NC3 for all NONL minimizers.

In all three cases, the resulting NC configuration is unique up to a global rotation and permutation of the class labels. This proves the theorem. $\qquad\square$

## B. Equivalence of SCL and prototype–softmax minimizers

Here we provide the proof of Theorem 4.2.

*Proof.* Fix $i \in [2M]$ with label $y_i$. Let $\mathcal{C}(i) = \{j \in [2M] : j \neq i, y_j = y_i\}$, $\mathcal{B}_c = \{j \in [2M] : y_j = c\}$, $n_c = |\mathcal{B}_c|$, and $\hat{\boldsymbol{\mu}}_c = \frac{1}{n_c} \sum_{j \in \mathcal{B}_c} \boldsymbol{a}_j$.

**(A) SCL lower bound.** By unfolding the SCL loss defined in Equation (2), the per-example loss term can be written as

$$\ell_i^{\text{SCL}} = -\frac{1}{|\mathcal{C}(i)|} \sum_{l \in \mathcal{C}(i)} \frac{\boldsymbol{a}_i^\top \boldsymbol{a}_l}{\tau} + \log \sum_{j \in [2M] \backslash \{i\}} \exp(\boldsymbol{a}_i^\top \boldsymbol{a}_j / \tau).$$

For the first term, using $\frac{1}{|\mathcal{C}(i)|} \sum_{l \in \mathcal{C}(i)} \boldsymbol{a}_l = \frac{n_{y_i} \hat{\boldsymbol{\mu}}_{y_i} - \boldsymbol{a}_i}{n_{y_i} - 1}$ and $\|\boldsymbol{a}_i\| = 1$ gives $-\frac{\boldsymbol{a}_i^\top}{\tau} \left( \frac{1}{|\mathcal{C}(i)|} \sum_{l \in \mathcal{C}(i)} \boldsymbol{a}_l \right) \geq -\frac{\boldsymbol{a}_i^\top \hat{\boldsymbol{\mu}}_{y_i}}{\tau}$.

For the second term, we group by class, subtract the self term and then apply Jensen classwise due to convexity of the exponential function:

$$\sum_{j \in [2M] \backslash \{i\}} e^{\boldsymbol{a}_i^\top \boldsymbol{a}_j / \tau} = \sum_{c=1}^K \sum_{l \in \mathcal{B}_c} e^{\boldsymbol{a}_i^\top \boldsymbol{a}_l / \tau} - e^{1/\tau} \geq \sum_{c=1}^K n_c \, e^{\boldsymbol{a}_i^\top \hat{\boldsymbol{\mu}}_c / \tau} - e^{1/\tau}.$$

Combining,

$$\ell_i^{\text{SCL}} \geq -\frac{\boldsymbol{a}_i^\top \hat{\boldsymbol{\mu}}_{y_i}}{\tau} + \log \left( \sum_{c=1}^K n_c \, e^{\boldsymbol{a}_i^\top \hat{\boldsymbol{\mu}}_c / \tau} - e^{1/\tau} \right) =: \ell_i^\star. \tag{13}$$

Equality in equation 13 holds iff every class-wise sum is collapsed, i.e., $\boldsymbol{a}_j = \hat{\boldsymbol{\mu}}_c$ for all $j \in \mathcal{B}_c$, because the positive-term bound is tight only when $\boldsymbol{a}_i^\top \hat{\boldsymbol{\mu}}_{y_i} = 1$ (so $\boldsymbol{a}_i = \hat{\boldsymbol{\mu}}_{y_i}$) and the classwise Jensen step is tight only when all within-class logits $\{\boldsymbol{a}_i^\top \boldsymbol{a}_l : l \in \mathcal{B}_c\}$ are equal.

**(B) Prototype loss lower bound.** Since $\boldsymbol{a}_i^\top \hat{\boldsymbol{\mu}}_{y_i} \leq 1$ for unit vectors, $e^{\boldsymbol{a}_i^\top \hat{\boldsymbol{\mu}}_{y_i} / \tau} \leq e^{1/\tau}$. Therefore

$$\underbrace{\sum_{c=1}^K n_c \, e^{\boldsymbol{a}_i^\top \hat{\boldsymbol{\mu}}_c / \tau} - e^{\boldsymbol{a}_i^\top \hat{\boldsymbol{\mu}}_{y_i} / \tau}}_{=:D_i^{\text{proto}}} \geq \underbrace{\sum_{c=1}^K n_c \, e^{\boldsymbol{a}_i^\top \hat{\boldsymbol{\mu}}_c / \tau} - e^{1/\tau}}_{=:D_i^\star},$$

and thus, with the *same* numerator $e^{\boldsymbol{a}_i^\top \hat{\boldsymbol{\mu}}_{y_i} / \tau}$,

$$\ell_i^{\text{proto}} = -\frac{\boldsymbol{a}_i^\top \hat{\boldsymbol{\mu}}_{y_i}}{\tau} + \log D_i^{\text{proto}} \geq -\frac{\boldsymbol{a}_i^\top \hat{\boldsymbol{\mu}}_{y_i}}{\tau} + \log D_i^\star = \ell_i^\star.$$

Averaging over $i$ gives the following inequalities for any batch $\boldsymbol{A}$:

$$L_{\text{SCL}}(\boldsymbol{A}) \geq L_\star(\boldsymbol{A})$$
$$L_{\text{proto}}(\boldsymbol{A}) \geq L_\star(\boldsymbol{A}).$$

**(C) Collapse–simplex makes all three equal.** By Graf et al. (2021, Theorem 2), any SCL global minimizer exhibits class-wise collapse, $\boldsymbol{a}_j = \boldsymbol{\zeta}_{y_j}$, and the directions $\{\boldsymbol{\zeta}_c\}$ form a centered regular $(K-1)$-simplex. Hence $\hat{\boldsymbol{\mu}}_c = \boldsymbol{\zeta}_c$ and $\boldsymbol{a}_i^\top \hat{\boldsymbol{\mu}}_{y_i} = 1$ for all $i$, making both inequalities above tight:

$$L_{\text{SCL}}(\boldsymbol{A}^\star) = L_\star(\boldsymbol{A}^\star) = L_{\text{proto}}(\boldsymbol{A}^\star).$$

Therefore $\min L_{\text{SCL}} = \min L_\star = \min L_{\text{proto}}$, all attained at the collapsed-simplex configurations.

**(D) Equality of argmin sets.** Let $\boldsymbol{A}$ minimize $L_{\text{proto}}$. Then $L_{\text{proto}}(\boldsymbol{A}) = \min L_{\text{proto}} = \min L_\star$, so $L_\star(\boldsymbol{A}) = L_{\text{proto}}(\boldsymbol{A})$, which forces $e^{\boldsymbol{a}_i^\top \hat{\boldsymbol{\mu}}_{y_i} / \tau} = e^{1/\tau}$ for every $i$, i.e., $\boldsymbol{a}_i^\top \hat{\boldsymbol{\mu}}_{y_i} = 1$ and hence $\boldsymbol{a}_i = \hat{\boldsymbol{\mu}}_{y_i}$ (class-wise collapse). Moreover $L_{\text{SCL}}(\boldsymbol{A}) = L_\star(\boldsymbol{A}) = \min L_{\text{SCL}}$, so $\boldsymbol{A}$ also minimizes SCL.

Graf's theorem then implies the class means form a centered simplex ETF. Thus the argmin sets of $L_{\text{SCL}}$ and $L_{\text{proto}}$ coincide (up to rotation and label permutation).

$\square$

## C. From Theory to Empirical Behavior

Theorem A.1 and Theorem 4.2, combined with the framing of our paper relative to the literature, imply specific behaviors that should manifest in practice. We organize this section around six such expected behaviors (EB1–EB6) and verify each empirically against the corresponding table or figure.

**EB1: Normalized losses converge to NC geometry faster than CE.**   Normalized losses make NC the unique global optimum on a benign strict-saddle landscape (Yaras et al., 2022), whereas CE admits unbounded rescaling that prevents NC even at its global optimum (Soudry et al., 2018).

*Validated:* Table 7 shows our methods reach CE-equivalent NC values in under $7.5\%$ of CE's iterations on 4/5 metrics. Figure 2 shows CE plateauing while our losses converge to $\geq 95\%$ NC (Table 3).

**EB2: NONL converges to NC faster than NormFace and NTCE.**   Decoupling alignment from uniformity removes competing gradient directions (Yeh et al., 2022), so NONL should follow a more direct optimization path.

*Validated:* Table 7 shows NONL converges $1.2$–$3.1\times$ faster than NormFace on the rank metrics.

**EB3: NTCE improves over NormFace.**   Expanding the negative set from $K$ prototypes to $M$ batch instances yields better contrastive objective estimates (Koromilas et al., 2024), strengthening inter-class separation.

*Validated:* Table 1 and Table 3 show NTCE outperforming NormFace on both accuracy and NC metrics.

**EB4: Class-mean prototypes from SCL are effective classifiers throughout training.**   Theorem 4.2 proves the minimizer sets of $\mathcal{L}_{\mathrm{SCL}}$ and $\mathcal{L}_{\mathrm{proto}}$ coincide, which means SCL already optimizes for classifier–feature alignment, not only at the exact (unreached) optimum.

*Validated:* Table 2 shows fixed prototypes match linear probing on 3/4 datasets and exceed it on ImageNet-100 by $+2.0\mathrm{pp}$, with a single forward pass replacing the full LP training phase.

**EB5: Classifier Learning with normalized losses and SCL reach the same NC geometry.**   Both are prototype-contrast methods on the hypersphere converging to the same simplex ETF.

*Validated:* In Table 3 both families arrive significantly closer to the NC structure compared to CE.

**EB6: Better NC geometry yields better downstream performance.**   The simplex ETF maximizes inter-class margins, which prior work links to improved generalization, transferability, and robustness (Bartlett et al., 2017; Ding et al., 2020; Fawzi et al., 2016; Galanti et al., 2021; Hein & Andriushchenko, 2017; Neyshabur et al., 2018; Papyan et al., 2020; Yang et al., 2022).

*Validated:* Tables 4 to 6 show consistent gains in transfer ($+5.5\%$ mean), long-tail (up to $+8.7\%$), and robustness (lower mCE).

**Where theory meets practical approximation.**   Our theorems characterize the geometry of global minimizers but do not analyze the optimization dynamics that reach them. In practice, our methods closely approach the theoretical NC optimum across all metrics (Table 3, Figure 2), substantially more so than CE or SCL with linear probing. The theory also does not characterize how batch size, learning-rate schedules, temperature $\tau$, or data augmentation affect convergence. For instance, our theorems hold for any $\tau > 0$, yet Figure 3 shows that temperature impacts finite-time performance in some cases; similarly, larger batches provide better approximations of the contrastive objective (Koromilas et al., 2024), confirmed by our Table 11. Understanding these optimization-level interactions remains an open direction. Our contribution is identifying the right target geometry and the losses that make it uniquely optimal, while the dynamics of reaching it deserve further study.

## D. Relation to Other Literature

This section expands on the related-work discussion in the main text, situating our framework relative to three additional bodies of work that our rebuttal discussion brought into focus: alternative routes to constraining radial freedom, the

associative-embedding line of work, and recent NC results under specific architectures.

### D.1. Relation to Graf et al. (2021) NC Results on Supervised Contrastive Learning

Our Theorem 4.2 strengthens the SCL minimizer characterization of Graf et al. (2021), which establishes that SCL minimizers exhibit NC1 (within-class collapse) and NC2 (simplex ETF of class means). It is natural to ask whether the existence of an optimal class-mean classifier follows trivially from NC1 + NC2; we show here that it does not, and that this matters in practice.

**Why the class-mean classifier does not follow trivially from NC1 + NC2.**   At the exact global minimum, where NC1 + NC2 are realized, the best linear classifier indeed has weight vectors corresponding to the class means; this much is straightforward. However, SCL, contrary to NTCE and NONL, which closely approximate NC ($\geq 95\%$ on all NC metrics in Table 3), does not in practice reach this exact minimum (Table 3: inter/intra erank $66.7/7.5$ on CIFAR-100 vs. the theoretical optimum $99/0$). When NC1 + NC2 is not realized, there is no prior theoretical basis to expect class-mean classifiers to work.

Theorem 4.2 fills this gap. It proves the minimizer-set equality $\arg \min \mathcal{L}_{\mathrm{SCL}} = \arg \min \mathcal{L}_{\mathrm{proto}}$, meaning SCL's objective is equivalent to optimizing the prototype-softmax classifier of Equation (6). That is, **the prototype classifier is the principled SCL classifier across the entire optimization trajectory, not only at the unreachable exact minimum**. This is why fixed prototypes (FP) work even at $\sim 95\%$ NC and why linear probing, which discards the projection head, returns to unnormalized features, and fits unconstrained weights and biases, is unnecessary.

**What is and is not proved by prior work.**   SCL contains no classifier weights in its formulation. Prior theorems therefore establish NC1 + NC2 for SCL minimizers but cannot address NC3 (classifier–feature alignment) or NC4 (zero bias), which are not even expressible without a classifier. Theorem 4.2 resolves this by proving that a classifier is already implicit in SCL: the class-mean prototypes *are* the classifier weights, enabling NC3 and NC4. Concretely, prior results characterize *what SCL solutions look like* (collapsed ETF); Theorem 4.2 proves *what SCL optimizes for* (prototype classification). It establishes that class-mean prototypes do not serve as post-hoc classifier weights but are the solution the SCL objective drives toward, making their direct use principled rather than heuristic.

**Practical consequence.**   The linear probing phase of SCL discards the projection head and trains on unnormalized encoder representations with unconstrained weights and biases. Fixed prototypes instead use the normalized representations from the projection head (the optimization space) and add no magnitude or bias. This rests on two contributions: (i) Theorem 4.2 proves the optimal classifier is already on the optimization space, and (ii) our normalized classifier framework, which enables classification on the normalized space rather than via post-hoc unnormalized probing.

### D.2. Alternative Approaches to Constraining Radial Freedom

There are several approaches to controlling radial degrees of freedom beyond normalized softmax: Riemannian optimization on the Stiefel manifold (Bansal et al., 2018; Huang et al., 2018), weight normalization (Salimans & Kingma, 2016), spectral normalization (Miyato et al., 2018), and layer rotation (Carbonnelle & De Vleeschouwer, 2019).

The critical observation is that all these methods are applied on top of standard CE, inheriting its fundamental limitations. Stiefel optimization constrains weight vectors to be mutually orthogonal (pairwise cosine 0) by definition, whereas the simplex ETF that NC requires has pairwise cosine $-1/(K-1)$, thus constituting a distinct geometric structure. Our normalized softmax can be understood as optimization on the oblique manifold (product of spheres), which Yaras et al. (2022) prove has a benign strict-saddle landscape specifically for NC. Weight and spectral normalization control magnitudes but still operate with unnormalized features and learnable biases. Layer rotation is a useful diagnostic but does not modify the training objective. In all cases, applying these constraints to CE still leaves: (i) additional regularization hyperparameters to eliminate radial freedom, (ii) no temperature to control the similarity distribution, and (iii) CE's fundamental loss-level limitations, *i.e.* small effective negative set and alignment–uniformity coupling, that no geometric constraint alone can resolve.

Normalized softmax addresses the constraint side cleanly by utilizing per-vector normalization on the right manifold, zero bias, and temperature control, while NTCE and NONL address the loss-level problems that persist even under perfect geometric constraints.

### D.3. Associative Embedding and Prototype-Augmented SCL

Our work also connects to the associative-embedding literature (De Brabandere et al., 2017; Neven et al., 2019; Newell et al., 2017), where embeddings are learned by contrasting in-group positives to out-of-group negatives and are applied to tasks where linear probing is not needed. While our work focuses on the principled SCL pipeline for classification, where linear probing is standard practice and where NC theory provides formal optimality guarantees, this conceptual link is meaningful and worth making explicit.

Concretely, eliminating linear probing is not relevant to associative methods (which already skip it), but our Theorem 4.2 is: it shows that optimizing per-instance alignment and uniformity (as in SCL) yields the same optima as optimizing with class-mean prototypes (as associative methods do). This gives theoretical room to apply SCL in associative method applications. However, associative methods operate in unconstrained Euclidean space where precisely the radial degree-of-freedom problem we identify prevents convergence to optimal geometry. Our hyperspherical framework addresses this: normalizing embeddings to the unit sphere eliminates scale ambiguity and provides geometric guarantees that associative methods lack. Our results suggest one could replace associative losses with SCL on the hypersphere, achieving the same optima more easily thanks to the benign loss landscape (Yaras et al., 2022).

Beyond SCL, NTCE and NONL open room for principled and more efficient CE-style training that uses learnable group embeddings (the classifier weights), which we leave as a natural direction for future work on associative embedding methods.

Closer in spirit, Gill et al. (2024) augment SCL with explicit prototype representations. Our perspective is complementary: rather than adding explicit prototypes to SCL, Theorem 4.2 proves that SCL already implicitly optimizes for them. The class-mean prototypes are not added on top of SCL; they are the classifier the SCL objective drives toward.

### D.4. NC under Specific Architectural and Training Choices

The standard NC theory builds on the unconstrained-features (UFM) and layer-peeled (LPM) models, which abstract away architectural details. Súkeník et al. (2024) and Súkeník et al. (2025) have begun to formally justify these idealizations for end-to-end training: proving deep UFM optimality for non-linear models and, in follow-up work, formally reducing end-to-end ResNet/Transformer training to an equivalent UFM, with the approximation tightening as depth grows. These results provide grounding for the UFM/LPM assumptions in Theorem A.1 and Theorem 4.2.

A second strand examines NC under specific activation functions. Kini et al. (2024) show that when using ReLU, the optima of SCL is an orthogonal frame with collapsed in-class representations, rather than a centered simplex ETF. This minimizer can be plugged directly into our proof (Section B, Part C and Step 3) in place of the Graf et al. minimizer. When changing the minimizer, the argmin equivalence from Theorem 4.2 still holds: $\arg\min \mathcal{L}_{\text{SCL}} = \arg\min \mathcal{L}_{\text{proto}}$ is structural and does not depend on which minimizer shape is realized. The geometric implications, however, do depend on the activation, and we acknowledge this in Section 5.4. We also note that, contrary to NTCE and NONL, SCL does not achieve $\geq 95\%$ NC in our experiments (Table 3), so the gap to the activation-dependent optimum remains a separate empirical concern.

A third thread examines the relationship between NC and transfer. Chen et al. (2022) and Kornblith et al. (2021) report a trade-off between collapse and downstream transfer in the coarse-to-fine setting, where collapsed representations may fail to preserve fine-grained sub-class structure within a coarse class. Harun et al. (2025) propose explicitly controlling the degree of NC at intermediate layers, with their method enforcing NC at the classification layer, which is the layer we operate on, and reporting strong transfer, validating that NC at the final layer is compatible with strong transfer in the standard large-to-small protocol of Khosla et al. (2020), which is the protocol we evaluate. The broader NC literature similarly establishes that collapsed, maximally separated representations improve generalization, robustness, and transfer in this standard protocol (Bartlett et al., 2017; Ding et al., 2020; Fawzi et al., 2016; Galanti et al., 2021; Hein & Andriushchenko, 2017; Khosla et al., 2020; Neyshabur et al., 2018; Papyan et al., 2020; Soudry et al., 2018). The coarse-to-fine regime is outside our evaluation scope; we list it in Section 5.4.

## E. Implementation Details

Experiments are conducted on four standard image classification datasets: *CIFAR10, CIFAR100, ImageNet-100, and ImageNet1K*, following common representation learning benchmarking practices (Khosla et al., 2020; Markou et al., 2024; Wang et al., 2021; Yeh et al., 2022). We use ResNet50 for ImageNet-100/ImageNet1K and ResNet18 for CIFAR10/CIFAR100. All models are trained using SGD optimizer for 500 epochs on ImageNet1K (temperature 0.1) and

ImageNet-100 (batch size 1024, temperature 0.2) and 1000 epochs on CIFAR10/CIFAR100. For ImageNet1k in order to enable fair comparison we report for each method its best accuracy for training with batch size 2048, 4096, and 8192. For CIFAR10/100 we set the batch size to 512 and evaluate all 11 temperatures in the set [0.07, 0.1, 0.2, 0.3, 0.4, 0.5, 0.6, 0.7, 0.8, 0.9, 1]. In Table 1 and Table 3 we report for each method the best performing temperature. For Supervised Contrastive Learning we perform the linear probing phase for the typical 90 epochs. Our code is publicly available at
https://github.com/pakoromilas/nc_by_design

### E.1. Classifier Learning Methods (CE, NormFace, NTCE, NONL)

For the family of classifier learning methods, we employ the following hyperparameters across datasets:

**CIFAR10/CIFAR100.** Models are trained for 1000 epochs with batch size 512. We use SGD optimizer with momentum 0.9, weight decay $10^{-4}$, and initial learning rate 0.2. The learning rate follows a cosine annealing schedule throughout training, decaying to a minimum value of $\eta_{\min} = \eta_0 \times 0.1^3$ where $\eta_0$ is the initial learning rate. Data augmentation consists of RandomResizedCrop with scale (0.2, 1.0), RandomHorizontalFlip, and standard normalization with dataset-specific mean and standard deviation values.

**ImageNet-100.** ResNet50 models are trained for 500 epochs with batch size 1024 (256 per GPU with 4 GPUs). We employ SGD optimizer with momentum 0.9, weight decay $10^{-4}$, and initial learning rate 0.1, which is automatically scaled based on the total batch size. We use cosine annealing scheduler with 10 epochs of linear warmup from 0.01 to the target learning rate. After warmup, the learning rate follows a cosine decay to $\eta_{\min} = \eta_0 \times 0.1^3$. Synchronized BatchNorm is enabled across GPUs. Data augmentation includes RandomResizedCrop(224) with scale (0.2, 1.0), RandomHorizontalFlip, and standard ImageNet normalization.

**ImageNet1K.** ResNet50 models are trained for 500 epochs with batch size 2048 (256 per GPU with 8 GPUs). Hyperparameters follow the same configuration as ImageNet-100, with SGD optimizer (momentum 0.9, weight decay $10^{-4}$), initial learning rate 0.1 with automatic scaling based on batch size. We apply 10 epochs of linear warmup followed by cosine annealing to $\eta_{\min} = \eta_0 \times 0.1^3$. Data augmentation and normalization follow ImageNet-100 settings.

### E.2. Supervised Contrastive Learning

For supervised contrastive methods, we implement a two-phase training procedure:

**Phase 1: Contrastive Training.**

**CIFAR10/CIFAR100:** Models are trained for 1000 epochs with batch size 512. SGD optimizer is used with momentum 0.9, weight decay $10^{-4}$, and initial learning rate 0.05. The learning rate follows cosine annealing schedule throughout training, decaying to $\eta_{\min} = \eta_0 \times 0.1^3$. We use extensive data augmentation including RandomResizedCrop with scale (0.2, 1.0), RandomHorizontalFlip, ColorJitter(0.4, 0.4, 0.4, 0.1) with probability 0.8, and RandomGrayscale with probability 0.2. Each image generates two augmented views for contrastive learning.

**ImageNet-100:** ResNet50 encoder with 128-dimensional projection head is trained for 500 epochs with batch size 1024. We use SGD optimizer with momentum 0.9, weight decay $10^{-4}$, and base learning rate 0.8 (automatically scaled by batch size). Learning rate follows cosine annealing with 10 epochs linear warmup from 0.01, then decays following a cosine schedule to $\eta_{\min} = \eta_0 \times 0.1^3$. Data augmentation extends CIFAR settings with the addition of Gaussian blur for ImageNet scale images.

**ImageNet1K:** Training spans 500 epochs with batch size 2048 using the same optimizer configuration as ImageNet-100. Base learning rate is set to 0.1 with automatic scaling. We employ cosine annealing with 5 epochs warmup from 0.01, followed by cosine decay to $\eta_{\min} = \eta_0 \times 0.1^3$. The same augmentation pipeline as ImageNet-100 is used.

**Phase 2: Linear Evaluation.** For all datasets, we freeze the learned encoder and train a linear classifier on top of the representations:

**CIFAR10/CIFAR100:** Linear classifier is trained for 100 epochs using SGD with learning rate 5.0, momentum 0.9, and zero weight decay. Learning rate is decayed by factor 0.2 at epochs 60, 75, 90 using a step scheduler.

**ImageNet-100:** Linear evaluation runs for 90 epochs with SGD optimizer, learning rate 2.0, momentum 0.9, and zero

*Table 7.* Convergence speed (% of training iters): **(I)** time to reach the 95% NC threshold; **(II)** time to match CE's final value; "0%" indicates the target is met at the first logged eval.

| Method | Instance alignment | Weight alignment | Weights erank | Intra erank | Inter erank |
|---|---|---|---|---|---|
| **(I) NC convergence to $95\%$ threshold (ratio to max iterations)** | | | | | |
| NormFace | 79.4% | 8.2% | 52.6% | 45.4% | 56.2% |
| NTCE | 79.4% | 6.8% | 56.4% | 36.6% | 52.4% |
| NONL | 79.4% | 7.4% | 34.6% | 14.6% | 47.2% |
| **(II) CE convergence to converged value (ratio to CE converged iteration)** | | | | | |
| NormFace | 2.2% | 2.0% | 66.3% | 0% | 7.4% |
| NTCE | 2.2% | 1.8% | 73.9% | 0% | 7.4% |
| NONL | 2.2% | 1.8% | 35.4% | 0% | 6.0% |

weight decay. Learning rate decay by factor 0.2 occurs at epochs 30, 60, 80 using a step scheduler.

**ImageNet1K:** Linear classifier training spans 90 epochs with SGD, learning rate 0.8, momentum 0.9, and zero weight decay. The same step decay schedule as ImageNet-100 is applied.

### E.3. Additional Implementation Details

For distributed training on ImageNet datasets, we employ DistributedDataParallel with one process per GPU. Random seed is fixed at 42 for reproducibility. The cosine annealing scheduler is implemented following the standard formulation: $\eta_t = \eta_{\min} + \frac{1}{2}(\eta_0 - \eta_{\min})(1 + \cos(\frac{\pi t}{T}))$, where $t$ is the current epoch and $T$ is the total number of epochs. For experiments with warmup, the warmup period linearly interpolates from the warmup starting learning rate to the initial learning rate before transitioning to cosine annealing. Temperature parameter $\tau$ is searched over the range [0.07, 0.1, 0.2, ..., 1.0] for CIFAR experiments, while ImageNet experiments use the optimal temperature found through preliminary experiments (0.1 for supervised contrastive, 0.2 for classifier learning methods). All models use standard weight initialization and no additional regularization beyond weight decay.

## F. Convergence Dynamics.

On CIFAR-100, we track NC metrics and define convergence as the earliest iteration where the exponentially-weighted moving average enters and remains within a metric-specific tolerance around the 95% NC threshold.

In Table 7(I) the convergence speed to 95% of theoretical NC thresholds is quantified. Normalized losses reach these thresholds, *typically early in training*. NONL converges faster with **gains over NormFace for the rank metrics** (1.2-3.1 speedup), benefiting from simplified optimization without competing terms. Table 7(II) benchmarks against CE's converged values. The acceleration is dramatic: normalized losses reach CE-equivalent values in under 7.5% of CE's required iterations across 4/5 metrics, while **NONL converges faster**. This demonstrates that normalized losses fundamentally restructure the optimization landscape, *enabling direct paths to neural collapse*.

In Figure 2 the training dynamics are demonstrated. While cross-entropy (CE) achieves perfect training accuracy, it fails to reach neural collapse geometry, plateauing at suboptimal metric values. CE's accuracy improvements appear to *derive solely from magnitude and bias adjustments* rather than geometric reorganization. In contrast, our methods *simultaneously optimize all NC metrics throughout training*, converging to proper NC geometry while maintaining optimal accuracy.

## G. Extra Ablation Studies

### G.1. Role of the Projection Head

In Table 8 we demonstrate the importance of the projection head in contrastive training. Across three datasets, except on the relatively simple CIFAR-10 benchmark, removing the head consistently reduces accuracy by more than 2 points. At first glance, one might expect the opposite: discarding the head should let the loss act directly on the final encoder embeddings on the unit hypersphere. We hypothesize that the projection head helps primarily by imposing a beneficial dimensionality bottleneck. With ResNet-50, the encoder's representation is 2048-dimensional, whereas the projection head maps it to

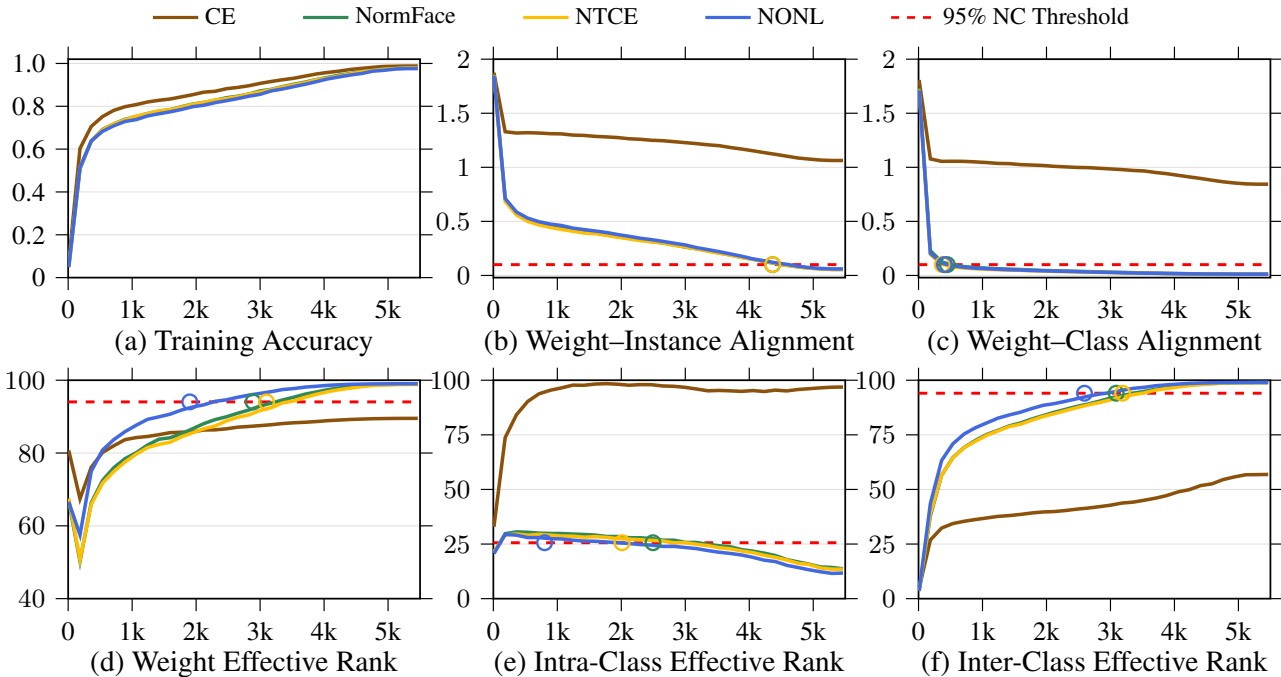

*Figure 2.* **NC convergence on CIFAR-100.** Six metrics vs. training iterations; red dashed lines mark the 95% NC threshold and circles denote each method's convergence.

*Table 8.* **Contrastive Learning Results - Without Projection Head.** Performance comparison across different classifier learning approaches without projection head.

| Classifier Learning | Loss | CIFAR-10 | CIFAR-100 | ImageNet-100 | ImageNet-1K |
|---|---|---|---|---|---|
| LINEAR PROBING | SCL | 95 | 70.6 | 84.1 | 71 |
| NORMALIZED LINEAR PROBING | SCL | 95 | 71.4 | 84.3 | 72.1 |
| FIXED PROTOTYPES | SCL | 95 | 71.4 | 84.7 | 70.1 |

128 dimensions. For a $K$-class problem (e.g., $K = 100$), the ideal equiangular tight frame (ETF) geometry lives in a $(K − 1)$-dimensional subspace. Encouraging embeddings to adopt this structure is plausibly easier in a 128-dimensional space than in a 2048-dimensional one, where the optimizer has many more irrelevant directions to explore.

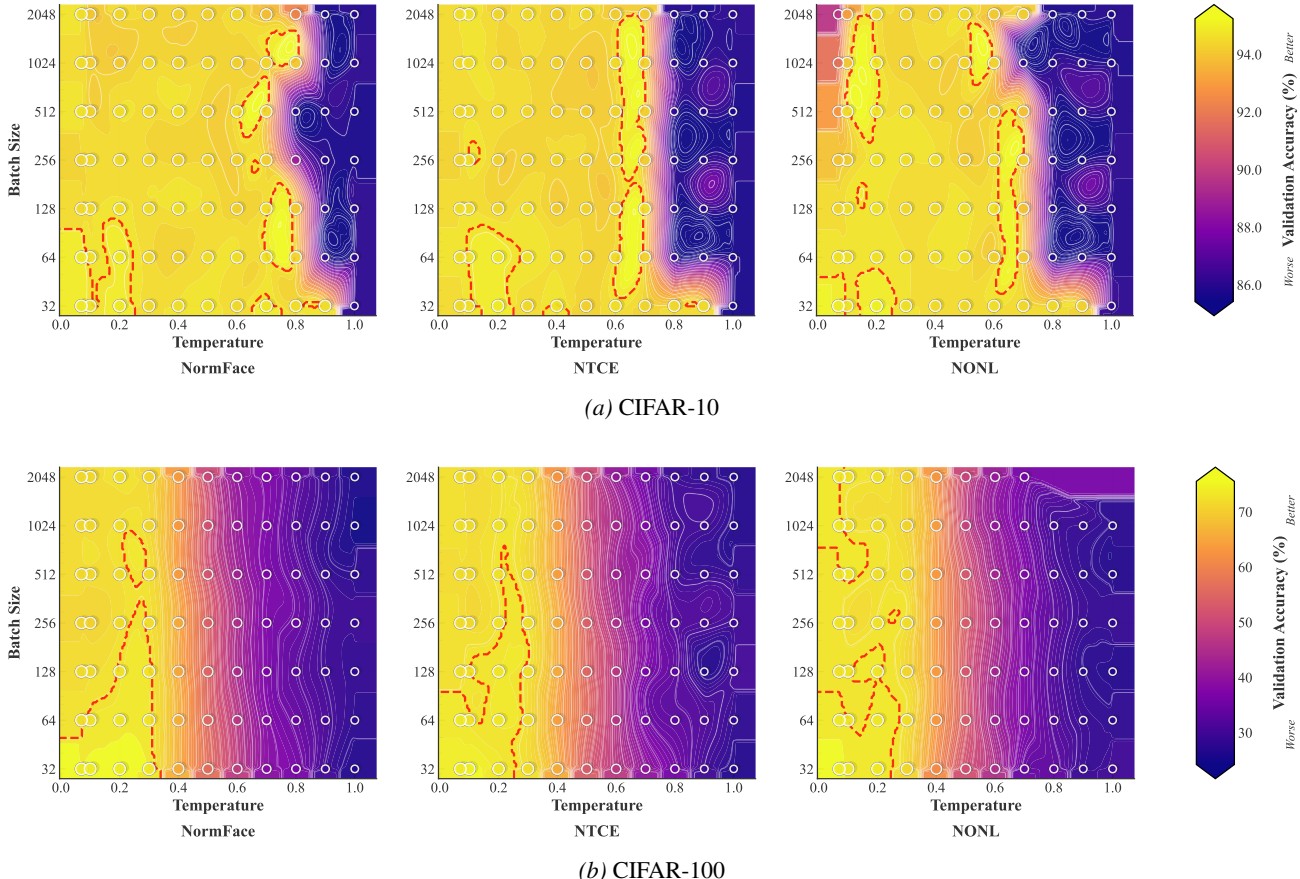

*(a)* CIFAR-10

*(b)* CIFAR-100

*Figure 3.* **Validation Accuracy (%) Phase Diagrams.** Classification accuracy on validation set. Higher values indicate better generalization performance. Each subplot shows the performance landscape across temperature and batch size hyperparameters for different loss functions: NormFace, NTCE, and NONL. Brighter regions indicate superior performance. White contour lines indicate iso-performance curves for detailed analysis. Red dashed contours highlight optimal parameter regions (top 10% performance). Scatter points represent individual experimental runs with performance-based sizing. Each dataset uses its own optimal colorbar range. Results originate from grid runs across temperature values in [0.07, 1.0] and batch sizes in 32, 64, 128, 256, 512, 1024, 2048.

## G.2. Effective hyperparameter ranges

Normalized softmax losses introduce two hyperparameters inherited from contrastive learning: the temperature $\tau$, which controls the sharpness of the similarity distribution, and the need for larger batch size $B$, which governs the number of in-batch negatives. We assess their impact by grid–searching $\tau \in [0.07, 1.0]$ (11 values) and $B \in \{32, 64, 128, 256, 512, 1024, 2048\}$ on CIFAR-10 and CIFAR-100 with NormFace, NTCE, and NONL.

Figure 3 shows consistent "sweet spots" across methods: accuracy forms a pronounced band at moderate temperatures, with performance degrading for overly large $\tau$ and, to a lesser extent, for very small $\tau$. The location of this band shifts toward slightly smaller temperatures as the number of classes increases (CIFAR-100 vs. CIFAR-10), mirroring observations in self-supervised contrastive learning (Chen et al., 2020). Within the effective temperature range, performance is comparatively insensitive to $B$, yielding a broad plateau over batch sizes—large batches can help, but are not strictly required.

In practice, these trends provide the same reliable defaults as in self-supervised contrastive learning (Chen et al., 2020): $\tau \in \{0.1, 0.2\}$ works well for small- to medium-class datasets, while $\tau \in \{0.07, 0.1\}$ is preferable for larger-class settings. Thus, although normalized softmax losses expose additional hyperparameters, their effective ranges are narrow and stable, so a small amount of tuning (or even these defaults) is typically sufficient to reach near-peak accuracy.

## G.3. Temperature Ablation at Batch Size 512

Tables 9 and 10 report validation accuracy across temperatures $\tau \in \{0.07, 0.1, 0.2, \ldots, 1.0\}$ at batch size 512, the configuration used for the CIFAR results in Table 1. **Bold** marks the temperature reported in Table 1 (i.e., the best $\tau$ per method). We observe that the optimal $\tau$ shifts toward smaller values as the number of classes grows, consistent with self-supervised contrastive learning (Chen et al., 2020).

*Table 9.* CIFAR-100 — Validation Accuracy (%) at batch size 512. **Bold** = value reported in Table 1 (best $\tau$ per method).

| Objective | $\tau$=0.07 | 0.1 | 0.2 | 0.3 | 0.4 | 0.5 | 0.6 | 0.7 | 0.8 | 0.9 | 1.0 |
|---|---|---|---|---|---|---|---|---|---|---|---|
| NormFace | 71.7 | 71.7 | 72.0 | **72.4** | 63.6 | 52.0 | 42.5 | 37.0 | 33.6 | 28.9 | 27.6 |
| NTCE | 72.0 | 71.7 | **72.9** | 71.4 | 64.0 | 52.3 | 42.7 | 38.6 | 30.2 | 29.0 | 28.2 |
| NONL | 73.2 | **73.6** | 72.5 | 71.6 | 63.4 | 52.6 | 42.8 | 37.2 | 31.2 | 28.2 | 26.8 |

*Table 10.* CIFAR-10 — Validation Accuracy (%) at batch size 512. **Bold** = value reported in Table 1 (best $\tau$ per method).

| Objective | $\tau$=0.07 | 0.1 | 0.2 | 0.3 | 0.4 | 0.5 | 0.6 | 0.7 | 0.8 | 0.9 | 1.0 |
|---|---|---|---|---|---|---|---|---|---|---|---|
| NormFace | 94.6 | 94.3 | 94.5 | 94.5 | 94.6 | 94.7 | **94.8** | 94.6 | 85.9 | 85.9 | 85.7 |
| NTCE | 94.2 | 94.3 | 94.3 | 94.4 | 94.5 | 94.6 | **94.7** | 94.5 | 86.2 | 85.9 | 85.7 |
| NONL | 92.9 | 94.7 | **94.9** | 94.6 | 94.4 | 94.3 | 94.1 | 94.3 | 86.0 | 85.9 | 85.9 |

## G.4. Need for large batch size

In Table 11 NONL benefits from larger batches, reaching +1.1% over CE on ImageNet-1K with an appropriate batch size. We explicitly acknowledge that needing larger batches is a practical drawback, but this is a typical limitation of contrastive methods (including SCL (Khosla et al., 2020) and many SSL methods (Chen et al., 2020)). Our work highlights that supervised normalized classifiers are already performing a kind of contrastive learning with class prototypes, and that fully reaping the generalization and robustness benefits naturally points toward larger effective negative sets. Recent SSL methods (memory banks, queues, momentum encoders) suggest promising ways to increase the number of negatives without linearly scaling batch size; we now emphasize this as a natural direction for future extensions of NONL.

## G.5. Applicability to larger architectures

In Table 12 we test using deeper and higher-capacity models, where we observe the same consistent trend: our normalized prototype-based losses (NTCE and NONL) improve upon CE in terms of accuracy while inducing significantly stronger NC geometry. These results indicate that the benefits of our objectives are not limited to small or medium-scale architectures, but extend to larger backbones as well.

## G.6. Statistical Robustness

ImageNet-1K runs require $\sim$200 GPU-hours each, and works of this scale typically report single runs (Chen et al., 2020; He et al., 2020; Khosla et al., 2020), allocating compute to scaling rather than repetition. To address concerns about statistical robustness we provide 5-seed results on the smaller datasets, with temperature fixed to the value reported in Table 1 (i.e., the best $\tau$ per method). Table 13 reports mean $\pm$ standard deviation. On CIFAR-100, NONL exceeds CE by roughly four standard deviations, and the relative ordering CE $<$ NormFace $<$ NTCE $<$ NONL is consistent at fixed configuration, validating the superiority of our methods. We note that CIFAR-10 with a ResNet-18 is near-saturated at $\sim$94.6%, leaving limited headroom for separation between methods on this benchmark; nevertheless NONL still leads.

On the role of temperature: $\tau$ does not grant normalized losses extra expressivity over CE. Standard CE can replicate the effect of any $\tau$ through weight magnitudes (setting $\|\boldsymbol{w}_c\| \propto 1/\tau$) and additionally has free biases and unconstrained feature norms, rendering it strictly more flexible. NormFace, which is not one of our proposed methods, already uses $\tau$ and already outperforms CE, isolating the benefit of temperature from our contributions; on ImageNet-1K we use a single typical $\tau$ without grid search (Section E), yet gains persist.

## G.7. Computational Cost of Linear Probing vs. Fixed Prototypes

The standard linear probing protocol (Chen et al., 2020; Khosla et al., 2020) applies data augmentation at every epoch, so representations change and cannot be cached; the full encoder forward pass must repeat $T$ times for $T$ probing epochs, giving

*Table 11.* Top-1 accuracy (%) on ImageNet-1K for different batch sizes and loss functions.

| Loss | 2048 | 4096 | 8192 |
|------|------|------|------|
| CE | 75.4 | 75.4 | 75.1 |
| NORMFACE | 75.6 | 76.4 | 76.3 |
| NTCE (ours) | **76.0** | **76.7** | **76.7** |
| NONL (ours) | 75.0 | 76.2 | 76.5 |

*Table 12.* ImageNet-100 top-1 accuracy (%) for different backbones. Best results per column are highlighted in **green**. The last row reports the relative improvement of NONL over CE.

| Method | ResNet-50 | ResNet-101 | ResNet-152 | Mean |
|--------|-----------|------------|------------|------|
| CE | 84.4 | 85.3 | 85.5 | **85.1** |
| NormFace | 84.4 | 85.4 | 85.6 | **85.1** |
| NTCE | 84.7 | 85.4 | 85.4 | **85.2** |
| NONL | **84.9** | **85.5** | **85.8** | **85.4** |
| $\Delta$(NONL$-$CE) | +0.6% | +0.2% | +0.4% | **+0.4%** |

$T \times N$ complexity. A single-epoch cache-and-train approach is possible but deviates from the established protocol and yields lower accuracy, since the LP classifier no longer sees the augmentation-induced variance that the standard protocol relies on. Table 14 reports the gap.

On 8×B200 GPUs, 500 epochs of pretraining take $\sim$ 25h ($\sim$ 200 GPU-hours), while 90 LP epochs add $\sim$ 3h ($\sim$ 24 GPU-hours), roughly $\sim$\$342 on AWS per run. FP requires exactly one forward pass per sample, eliminating this overhead while matching LP accuracy (Table 2).

*Table 13.* Mean $\pm$ standard deviation of validation accuracy (%) across 5 seeds, with temperature fixed to the value reported in Table 1.

| Method | CIFAR-100 | CIFAR-10 |
|---|---|---|
| CE | $72.16 \pm 0.20$ | $94.52 \pm 0.15$ |
| NormFace | $72.32 \pm 0.17$ | $94.64 \pm 0.10$ |
| NTCE | $72.78 \pm 0.21$ | $94.68 \pm 0.07$ |
| NONL | $\mathbf{73.38 \pm 0.28}$ | $\mathbf{94.84 \pm 0.15}$ |

*Table 14.* Effect of caching encoder features during linear probing on SCL representations. Standard LP performs augmentation at every epoch (requiring $T \times N$ forward passes); cached LP computes encoder features once and trains a classifier on the cached representations ($N$ forward passes total but no augmentation during LP). FP eliminates LP entirely.

| | Standard LP | Cached LP | FP (ours) |
|---|---|---|---|
| CIFAR-100 | 73.9 | 72.3 | 73.9 |
| CIFAR-10 | 95.0 | 94.4 | 95.0 |

