# OpenReview forum: "Neural Collapse by Design: Learning Class Prototypes on the Hypersphere"
_ICML.cc/2026/Conference — ICML 2026 regular_

### Official Review · Reviewer_8bKP · 2026-02-27

**Soundness:** 1
**Presentation:** 3
**Significance:** 3
**Originality:** 3
**Overall Recommendation:** 3
**Confidence:** 4

**Summary:**

This paper analyzes the relative geometry between representations at the final linear layer of neural networks trained for supervised classification in two settings.

In a first part, two modifications to the cross-entropy loss are presented. First (NTCE), the cross entropy is not computed over inner products between representations and weight vectors, where the representation is fixed and the weight vectors vary. Instead one selects the weight vector of the class of some representation and then computes cross entropy with respect to this fixed weight vectors and varying representations (i.e. a vector of dimension = batch size). Second (NONL), the denominator of the cross entropy is only computed over weight-representation pairs from different classes. Theorem 4.1 then shows that these modifications to cross-entropy lead to neural collapse at minimal loss.
In a second part, it is shown that supervised contrastive learning (SCL) can be understood as a prototype based method, where prototypes correspond to centers of mass of classwise representations, leading to a variant of SCL without training the linear classifier after disposing the projection head once the encoder is trained.

Empirically, all 3 proposed modifications are evaluated in terms of classification performance on standard vision benchmark datasets (CIFAR and Imagenet). The degree to which models trained with these loss functions achieve neural collapse on the CIFAR-10 & CIFAR-100 datasets is measured. Moreover, transfer learning capabilities, robustness to corruptions and class imbalances are studied.

**Compliance With Llm Reviewing Policy:**

Affirmed.

**Final Justification:**

Over two rounds of author responses (the second one being closer to the evidence and therefore far more productive) most of my points have been addressed. I am certain that the proposed changes and additional reproducibility information will improve the manuscript, as will the clarifications on relation to prior work, and based on this, I raise my score.

I am still not convinced about the novelty of one theoretical contribution stated in the abstract (see second response), but as this is only a part of this work, it does not prohibit acceptance.

Overall, I rate this submission as borderline, somewhere between weak reject and weak accept with the exact rating depending on how willing the authors are to tone down the writing so that it reflects the conclusions that can be drawn (and the ones that cannot) from the paper more accurately.

**Key Questions For Authors:**

- Would you clarify how your theoretical results relate to (Graf et al., Dissecting Supervised Contrastive Learning, ICML 2021) and (Koromilas et al., Bridging Mini-Batch and Asymptotic Analysis in Contrastive Learning: From InfoNCE to Kernel-Based Losses, ICML 2024). Where and how do you use their results in your proofs? Which parts of your results are novel?
For example, it seems to me that (see abstract) *"that supervised contrastive learning already produces an optimal classifier during training, the prototype classifier whose weights are given by class-wise feature means..."* is a direct consequence of neural collapse as shown in (Graf et al., Theorem 2).
- What are the training times for the linear probing step, and how does it relate to the total training time of the contrastive model?

**Limitations:**

yes

**Strengths And Weaknesses:**

## Soundness
**Theory.** The two theorems presented in this work are well-supported by formal proofs. While I did not check each step thoroughly, I did not find any errors so far.

**Experiments.** I am not satisfied by the empirical evaluation presented in this work. On the positive side, multiple experiments assessing a wide range of properties (classification performance, neural collapse quantification, transfer learning, long-tails classification, robustness to data corruptions) are performed. However, on the negative side, there are several issues that raise question about the validity of the results.
1. Hyperparameter selection (i). According to Section C in the appendix, hyperparameters of the newly introduced methods are evaluated over a grid (batch size x temperature) and the best testing performance is reported. Assuming that the baseline hyperparameters are selected from the same grid, this gives methods that don't come with a temperature parameter a systematic disadvantage, as they have less competitors.
2. Hyperparameter selection (ii). The selection process for the remaining hyperparameters (e.g. learning rate, schedule, number of epochs, etc.) is not specified. These parameters are the same for all methods and might have been selected based on the testing performance of NTCE, NONL or FP.
3. Statistical robustness. Often times, scores achieved by the different methods are similar (especially for the contrastive methods), but spread information (e.g. with respect to random initialization or differing training/testing splits) are not reported. This prohibits assessing whether the gains are significant.
4. Performance. There are no improvements over standard supervised contrastive learning. While the classification performance (tables 1 and 2) of NTCE and NONL methods seems (see above) to be better than the cross-entropy competitors, it is below that of standard supervised contrastive learning (LP), which is tied with the proposed FP method. On the other tasks, (robustness, long-tailed data) there is no comparison with supervised contrastive learning.
5. It is stated that the methods "eliminate hours of post-hoc classifier training", by which the linear probing step of contrastive learning is meant. However, training times are not reported. Similarly, it is stated that "Fixed prototypes [... require] only N forward passes versus T × N for training-based methods, where T is the number of epochs." Unless I am missing something, this appears wrong to me. The linear probing step requires a single forward pass through the entire model to get all representations. The representations can then be saved and reused to train a cheap logistic regression model.
6. The transfer learning experiments are introduced as "We first ask whether representations that lie closer to the NC regime are more generalizable to unseen tasks." But this is not analyzed in the experiment, which instead just compares the transfer learning performance of the different methods.
7. A key assumption to NTCE is that contrastive objectives need very large numbers of negatives to converge, for which the authors state that this is very-well investigated and refer to (He et al., Momentum Contrast for Unsupervised Visual Representation Learning, CVPR 2020). However, most works, including He et al., consider the semi-supervised setting. In the supervised setting, this is not as well-investigated. The varying batch size experiments in (Khosla et al., NeurIPS 2020) support the claim though.

**Introduction.** The introduction ends with a list of claims about the contributions of this work. However, the paper does not present sufficient empirical evidence that the presented methods "accelerate convergence", are "interpretable" (resp. more interpretable than the baselines), that they "realize their optimal NC structure" (in practice) or that they come with "faster training".

## Presentation
Overall, the manuscript is well-written. However, on my first read the relation between the different methods (NTCE, NONL) vs. FP was a unclear to me. This was because from the abstract I had the impression that the NTCE and NONL methods are modifications to supervised contrastive learning, which benefit from the insight that "supervised contrastive learning already produces an optimal classifier during training". However, both are direct adaptions of the cross entropy loss and are presented *before* the contrastive learning parts.

## Significance
This work presents modifications to well-established training procedures for classification models. Its significance mainly depends on the benefit of the modifications. Given the presented empirical evidence, I currently cannot assess this part (see soundness).
That being said, avoiding the linear probing step in supervised contrastive learning without compromising performance would be quite beneficial to practitioners.

## Originality
- The contribution of this paper and how it relates to existing work need to be fleshed out more clearly. See questions.
- Strongly related work is not cited. (i) (Kini et al., Symmetric Neural-Collapse Representations with Supervised Contrastive Loss: The Impact of ReLU and Batching, ICLR 2024) who show that representations obtained by supervised contrastive learning converge to an **orthogonal** frame if ReLU activations are used (as in the ResNet architectures that are used for the experiments). Notably, this result holds irrespective of class imbalances.
(ii) Gill et al., Engineering the Neural Collapse Geometry of Supervised-Contrastive Loss, arxiv, 2023, who compare supervised contrastive learning with additional prototype representations with cross entropy training.
- Please cite (Deng et al., ImageNet: A large-scale hierarchical image database, CVPR 2009), (Krizhevsky, Learning Multiple Layers of Features from Tiny Images), (He et al., Deep Residual Learning for Image Recognition, CVPR 2016) and the additional 6 datasets used for transfer learning.

---

> ### Author Rebuttal · Authors · 2026-03-29
>
> ## Empirical Validation
>
> Regarding **statistical robustness**: ImageNet-1K runs require ~200 GPU-hours each; papers at this scale report single runs (Khosla et al. 2020; Chen et al. 2020; He et al. 2020), allocating compute to scaling rather than repetition. We provide **new 5-seed results** on smaller datasets that validate the superiority of our methods:
>
> | Method | CIFAR-100 | CIFAR-10 |
> | --- | --- | --- |
> | CE  | 72.16±0.20 | 94.52±0.15 |
> | NormFace | 72.32±0.17 | 94.64±0.10 |
> | NTCE | 72.78±0.21 | 94.68±0.07 |
> | NONL | 73.38±0.28 | 94.84±0.15 |
>
> **Shared hyperparameters (batch size, lr, schedule, weight decay, epochs) are identical across all methods within each learning family**. CL methods share one configuration (Sec. C.1), SCL methods share another (Sec. C.2). The training pipeline (SGD with momentum, cosine annealing, standard augmentation) is from the established SupCon codebase (https://github.com/HobbitLong/SupContrast), whose default **hyperparameters were developed for CE**, if anything, this favors the baseline. CE, NormFace, NTCE, and NONL all **belong to the same functional family**, i.e., log-ratio of exponentials of dot products, so **identical optimization settings constitute a controlled comparison** where the loss is the only independent variable, following **standard practice** (Yeh et al. 2022; Koromilas et al. 2024).
>
> **On temperature**, we first note that $\tau$ does not grant normalized losses extra expressivity over CE. Standard **CE can replicate the effect of any $\tau$** through weight magnitudes (setting $|w_c| \propto 1/\tau$) and additionally has free biases and unconstrained feature norms rendering it  strictly more flexible. Given that, we interpret this concern as asking whether our gains stem from temperature tuning rather than methodological contributions. To directly address this: (a) NormFace, which is not our method, also uses $\tau$ and already outperforms CE, **isolating the benefit of temperature from our contributions**; (b) on ImageNet-1K, we use a single typical $\tau$ **without grid search** (Sec. C), yet gains persist; (c) our new **5-seed results** above show gains exceeding two standard deviations at a fixed $(\tau, B)$ configuration.
>
>
> ## NTCE and NONL vs SCL
>
> *On ImageNet-1K, both our objectives beat SCL*, NTCE with +1.6pp and NONL +1.4pp. However, **NTCE and NONL are not alternatives to SCL, they are alternatives to CE** within the standard classification pipeline (single view, no projection head, no LP). Within this pipeline, they outperform CE by +1.3pp/+0.9pp on IN-1K. SCL+FP addresses a different question: whether LP can be eliminated for SCL practitioners. We will make this distinction sharper in the text.
>
> Cross-family comparisons are not emphasized because pipelines differ fundamentally (SCL requires heavy augmentation, two views, projection head, LP). Robustness/long-tail comparisons are within the CL family to isolate the loss from augmentation effects.
>
> ## Training times
>
> The **standard LP protocol** (Khosla et al., 2020; Chen et al., 2020) **applies data augmentation at every epoch**, so representations change and cannot be cached and the full encoder forward pass must repeat $T$ times, giving $T \times N$ complexity. A single-epoch cache-and-train approach is possible but deviates from the established protocol and yields lower accuracy. On 8× B200 GPUs, 500 epochs of pretraining takes ~25h (200 GPU-hours), while **90 LP epochs add ~3h (24 GPU-hours)**. FP requires exactly one forward pass, **saving ~24 GPU-hours**.
>
> ## Relation to Graf et al. and Koromilas et al.
>
> We refer the reviewer to **our answer to reviewer WY8e regarding SCL+FP novelty and C3**. More briefly, Graf et al. prove SCL minimizers exhibit NC1+NC2 but **not NC3 or NC4**. NC3 is not trivially implied: SCL contains no classifier weights, so **NC3 is not even stateable in their framework**. Our Theorem 4.2 proves what SCL *optimizes for*, i.e. $argmin\{L_{SCL}\} = argmin\{L_{proto}\}$ (Eq. 6), not just what solutions *look like* (ie ETF). This reveals that **SCL actually includes a classifier in its formulation** and **enables FP to replace LP, a practical consequence no prior theorem supports.** Koromilas et al.'s $L_a/L_c$ characterization is used in one proof step; the proof structure and NTCE/NONL are novel. Koromilas et al. work on the SSL setup with no classes and show the losses converge to uniformity instead of ETF; Gill et al. add explicit prototypes to SCL; our theorem shows SCL already implicitly learns them. Kini et al. study ReLU effects on collapse. Dataset citations added.
>
> ## Introduction claims
> Our claims are supported by our theory and experiments:
> - "accelerate convergence" by Table 7/Figure 1 (<7.5% of CE's iterations)
> -  "realize NC" by Table 3 (≥95%)
> - "faster training" by LP elimination and faster convergence
> - we will revise "interpretable" to "geometrically interpretable" since ETF is a dictionary of interpretable directions.

---

> > ### Author Rebuttal · Reviewer_8bKP · 2026-04-02
> >
> > Thank you for the response. Some points have been addressed, but others have not.
> >
> > ### Soundness
> >
> > 1. Hyperparameter selection (1). I consider this partially adressed. Please report the accurracies at differing temperature parameters on CIFAR 10/100.
> >
> > 2. Hyperparameter selection (2). Please specify the hyperparameter selection process. Are you using the default pipeline and hyperparameter values from https://github.com/HobbitLong/SupContrast?
> >
> > 3. Statistical robustness. Thank you for the 5-seed results. Are these results again at the best performing temperature, or did you use a fixed temperature here, given the short time for rebuttal? Would you agree that wrt to seed selection (and potentially accounting for hyperparameter selection) that the gains on CIFAR-10 are significant, but on CIFAR-100 they are not?
> >
> > 4. Performance. Would you please expand on why comparing NTCE and NONL with supervised contrastive learning is not emphasized?
> >
> > 5. "eliminate hours of post-hoc classifier training". Thank you for providing numbers here. So linear probing takes around 10% of training time.
> > "A single-epoch cache-and-train approach is possible but deviates from the established protocol and yields lower accuracy." Which accuracy did you get?
> >
> > 6. not adressed
> > 7. not adressed, but I assume that the authors will adapt the writing.
> >
> >
> > ### Presentation. Not adressed. Will changes be made?
> >
> >
> > ### Related work
> > 1. I might have misunderstood your argument. If NC1 and NC2 are fulfilled, isn't it trivial that the best linear classifier has weight vectors corresponding to the class means?
> > 2. Would you clarify where and how you use the results of Koromilas et al. and Graf et al.?
> > 3. Do the results from Kini et al., who show SCL does not converge to an ETF when using ReLUS, affect your results or their conclusions?
> >
> > ### Claim in introduction
> > - accelerate convergence. Please clarify in the manuscript that you are referring to convergence to NC, not a reduction of necessary training time to achieve the same accuracy.
> > - realize NC. Achieving scores ≥95% does not imply realizing NC, 100% would.
> > - faster training. I am confused by your answer. Does faster convergence does mean reduction of training time? If so, then this is not evaluated in table 7.

---

> > > ### Author Response · Authors · 2026-04-05
> > >
> > > We thank for the continued engagement and address the follow-up points below.
> > >
> > > ## Soundness
> > >
> > > **S1.** Ablation at https://anonymous.4open.science/r/Neural_Collapse_rebuttal-5696/temperature_ablation.pdf
> > >
> > > **S2.** We use this codebase with the default pipelines and hyperparameters for the CIFAR datasets. Since it does not include hyperparams for ImageNet (see main_supcon.py & main_ce.py) we keep the pipeline and adapt the hyperparameters to ImageNet by sourcing from the SupCon paper, for example ResNet50 @ 500 epochs (Fig. 4c), batch 2048 (Fig. 4b), and temperature 0.1. For CIFAR temperature we ablate in Fig. 2. For learning rate we use the base LR and scale to different batch sizes using the linear scaling rule (Goyal et al., 2017). All hyperparameters are shared identically across methods within each family × dataset combination.
> > >
> > > To help results reproducibility and adoption of our methods we'll release the SupCon codebase augmented with our methods and ImageNet integration
> > >
> > > **S3.** Temperature is fixed to the ones used in Table 1
> > >
> > > On CIFAR100, mean +1.22pp is significant. The reviewer likely refers to CIFAR10, where we agree that this dataset is not suitable for demonstrating a method's superiority since it is relatively easy with 10 classes at ~94.6% leaving limited headroom for a ResNet18. It is however a standard benchmark in the literature. Beyond that, we report results on three demanding datasets, where the results consistently indicate performance gains. On CIFAR-10 we still get +0.3 over CE across 5 seeds.
> > >
> > > **S4.** Cross-family pipelines differ beyond the loss: compared to CL methods, SCL further uses two views, heavy augmentation, projection head, and LP. Comparing across families, we cannot isolate gains to the pipeline or the loss. If we only compare result-wise, on IN-1K NTCE/NONL outperform SCL+LP by +1.6/+1.4pp.
> > >
> > > **S5.**
> > > To clarify, 10\% of training is **~$342 on AWS per run** (~24 GPU-hours on B200), which FP eliminates. Below are the requested results validating the typical LP has superior performance:
> > >
> > > | | LP | Cached LP |
> > > |---|---|---|
> > > | CIFAR-100 | 73.9 | 72.3 |
> > > | CIFAR-10 | 95.0 | 94.4 |
> > >
> > > **S6.** the relative ordering of methods on NC metrics is consistent across datasets (Tables 3: CE ≪ NormFace < NTCE ≤ NONL), and the same ordering holds for transfer (Table 4).
> > >
> > > **S7.** Supervised evidence: Khosla et al. (2020) Fig. 4b, our Table 9. Will adapt writing
> > >
> > > ## Presentation
> > > We refer to our answer to is1c where we state the restructure of Section 4
> > >
> > > ## Related Work
> > >
> > > **Q1**. We thank for this comment which help us increase text's clarity.
> > >
> > > The reviewer suggests that if NC1+NC2 hold, the class-mean classifier follows trivially. We agree this would be straightforward at the exact global minimum. But SCL, contrary to NTCE/NONL getting  ≥95% NC, does not reach that minimum in practice (Table 3: inter/intra erank 66.7/7.5 vs. optimum 99/0). *When NC1 +NC2 is not realized, there is no prior theoretical basis to expect class-mean classifiers to work*.
> > >
> > > Thm 4.2 fills this gap. It proves argmin L_SCL = argmin L_proto, meaning SCL’s objective is equivalent to optimizing a prototype-softmax classifier (Eq. 6). That is **the prototype classifier is the principled SCL classifier across training, not only at the unreachable exact minimum.** This is why FP works apart from NC  and why LP, which discards the projection head, returns to unnormalized features, and fits unconstrained weights and biases, is unnecessary.
> > >
> > >
> > > **Q2.**
> > > Theorem A.1:
> > > - Step 1 reduces to class means via Jensen
> > > - Step 2 translates losses from Step 1 to Koromilas formulation and use their Thm 4.1
> > > - Step 3 lifts to sample level via tightness
> > >
> > > Theorem 4.2:
> > > - Part A constructs the SCL lower bound
> > > - Part B L_proto shares the same lower bound
> > > - Part C at any SCL minimizer both bounds are tight  (applied SCL optimizer exhibits class-wise collapse from Graf)
> > > - Part D proves the reverse: any L_proto minimizer also minimizes L_SCL
> > > - this minimizer is the one identified by Graf
> > >
> > > **Q3.** They show that when using ReLU the optima is an orthogonal frame with collapsed in-class representations. This minimizer can be easily plugged in our proof (see Q2, last step \& part C) instead of the Graf minimizer. *When changing the minimizer, the argmin equivalence from Thm 4.2 still holds*. We also report  (see Q1)  that, contrary to NTCE/NONL, SCL doesn't achieve  ≥95% NC. Will discuss this in the updated text
> > >
> > > ## Introduction Claims
> > >
> > > *Accelerate convergence* Will clarify this refers to NC convergence
> > >
> > >
> > > *Realize NC* Will reframe to “achieve ≥95% on all NC metrics.” To quantify: on CIFAR100, CE reaches 0.83 alignment and 96.4 inter erank; our methods reach 0.01 and 99 (optima: 0 and 99). This is the difference between NC not emerging and NC being closely approximated across metrics.
> > >
> > >
> > > *Faster training* Will clarify it refers to SCL where LP elimination saves ~24 GPU-hours. In CL, though faster NC convergence may enable reduced training in future works

---

### Official Review · Reviewer_5sH1 · 2026-03-05

**Soundness:** 3
**Presentation:** 3
**Significance:** 3
**Originality:** 3
**Overall Recommendation:** 4
**Confidence:** 3

**Summary:**

This paper studies how to make supervised learning more directly realize Neural Collapse (NC) in practice. The key idea is to view both normalized softmax classifier learning and supervised contrastive learning (SCL) as prototype-based learning on the unit hypersphere. Based on this perspective, the paper proposes two normalized objectives, NTCE and NONL, intended to improve convergence to NC by increasing the effective number of negatives and decoupling attraction from repulsion. The paper also argues theoretically that SCL already learns an optimal classifier in the form of class-mean prototypes, making a separate linear probing stage unnecessary, and empirically evaluates the proposed view and losses on CIFAR-10/100, ImageNet-100, and ImageNet-1K, along with transfer learning, long-tailed classification, and corruption robustness.

**Compliance With Llm Reviewing Policy:**

Affirmed.

**Key Questions For Authors:**

The strongest theoretical claims rely on idealized assumptions (balanced labels, UFM/LPM-style analysis, unit-norm representations/classifiers, and global minimizers), while the main empirical setting is finite-time deep network optimization. The paper makes a compelling conceptual bridge, but the gap between theorem and practical claim remains noticeable. I would have liked a clearer discussion of what exactly should or should not be expected to transfer from the theory to realistic deep models.

**Limitations:**

yes

**Strengths And Weaknesses:**

1. The paper has a clear and ambitious conceptual contribution: it unifies normalized softmax classification and supervised contrastive learning under a common hyperspherical prototype-contrast view. This is a clean perspective that helps connect several previously somewhat separate threads of work on NC, normalized classifiers, and contrastive learning.
2. Theoretical contributions are nontrivial. In particular, the paper proves NC optimality of the normalized losses under the balanced UFM/LPM setting, and it further establishes an equivalence between SCL and a prototype-softmax objective at the level of global minimizers. Even if some assumptions are idealized, these results are intellectually meaningful and help formalize the proposed unification.
3. The empirical evaluation is broad. Beyond standard top-1 accuracy, the paper evaluates several NC metrics, convergence speed, transfer learning, long-tailed classification, and ImageNet-C robustness. This breadth strengthens the paper’s practical narrative.
4. The fixed-prototype replacement for linear probing after SCL is practically appealing. Matching linear probing on most datasets while avoiding an additional training phase is a useful systems-level simplification.

---

> ### Author Rebuttal · Authors · 2026-03-29
>
> We thank Reviewer 5sH1 for the positive assessment and for recognizing the clean conceptual contribution, the nontrivial theoretical results, the broad empirical evaluation, and the practical appeal of fixed-prototype classification. We agree that the connection between theory and practice deserves explicit discussion and will add a dedicated paragraph in the experiments section.
>
> Below we discuss (A) the specific practical behaviors our theory predicts and their empirical confirmation, and (B) how the idealized assumptions relate to realistic deep network training.
>
> ## A. Expected behavior (EB) from theory and empirical validation
>
> Our theorems, combined with the framing of our paper relative to the literature, imply specific behaviors that should manifest in practice:
>
> **EB1: Normalized losses should converge to NC geometry faster than CE**, because they make NC the unique global optimum on a benign strict-saddle landscape (Yaras et al. 2022), whereas CE admits unbounded rescaling that prevents NC even at its global optimum (Soudry et al. 2018). *Validated:* in Table 7 our methods reach CE-equivalent NC in <7.5% of CE's iterations on 4/5 metrics. Figure 1 shows CE plateauing while our losses converge to ≥95% NC (Table 3).
>
> **EB2: NONL should converge to NC faster than NormFace and NTCE**, because decoupling alignment from uniformity removes competing gradient directions. *Validated:* in Table 7 NONL converges 1.2–3.1× faster than NormFace on rank metrics.
>
> **EB3: NTCE should improve over NormFace**, because expanding the negative set from $K$ prototypes to $M$ batch instances yields better contrastive objective estimates (Koromilas et al. 2024). *Validated:* in Tables 1, 3 NTCE outperforms NormFace on both accuracy and NC metrics.
>
> **EB4: Class-mean prototypes from SCL should be effective classifiers**, because Theorem 4.2 proves the minimizer sets of $L_{SCL}$ and $L_{proto}$ coincide — SCL already optimizes for classifier-feature alignment. *Validated:* in Table 2 FP matches LP on 3/4 datasets, exceeds it on IN-100 by +2.0pp, with a single forward pass.
>
> **EB5: Classifier Learning with normalized losses and SCL should reach the same NC geometry**, because both are prototype-contrast methods on the hypersphere converging to the same simplex ETF. *Validated:* in Table 3 both families are arrive significantly closer to the NC structure compared to CE.
>
> **EB6: Better NC geometry should yield better downstream performance**, because the simplex ETF maximizes inter-class margins. *Validated:* in Tables 4–6 consistent gains in transfer (+5.5%), long-tail (up to +8.7%), and robustness (lower mCE).
>
> **Where theory meets practical approximation.** Our theorems characterize the geometry of global minimizers but **do not analyze the optimization dynamics** that reach them. In practice, our methods closely approach the theoretical NC optimum across all metrics (Table 3, Figure 1), substantially more so than CE or SCL with linear probing. The theory also does not characterize how batch size, learning rate schedules, temperature $\tau$, and data augmentation affect convergence. For instance, our theorems hold for any $\tau > 0$, yet Figure 2 shows temperature impacts finite-time performance in some cases; similarly, larger batches provide better approximations of the contrastive objective (Koromilas et al. 2024), confirmed by our Table 9. Understanding these optimization-level interactions remains an open direction. Our contribution is identifying the right target geometry and the losses that make it uniquely optimal, while the dynamics of reaching it deserve further study.
>
> ## B. On the theoretical assumptions
>
> The reviewer identifies four:
>
> **(i) Balanced labels.** NC-like structure remains optimal under imbalance, with geometry generalizing to sample-size-dependent angles (Hong & Ling, JMLR 2024; Thrampoulidis et al., NeurIPS 2022). We assume balance for cleaner theorems, but Table 6 confirms gains under severe imbalance.
>
> **(ii) UFM/LPM.** The standard framework across NC literature. Súkeník et al. have formally justified it: proving deep UFM optimality for non-linear models (NeurIPS 2024) and, in follow-up work (NeurIPS 2025), formally reducing end-to-end ResNet/Transformer training to an equivalent UFM, with the approximation tightening as depth grows.
>
> **(iii) Unit-norm constraints.** A deliberate practical choice: without normalization, CE admits spurious local minima and unbounded rescaling (Soudry et al. 2018). With normalization on the oblique manifold, Yaras et al. (2022) prove every local minimum is global. Validated by our NC convergence vs. CE's failure (Table 3).
>
> **(iv) Global minimizers.** The benign strict-saddle landscape means no spurious local minima, which translates to SGD better approximating near-global solutions, evidenced by our NC metrics and fast convergence (Table 7).

---

> > ### Author Rebuttal · Reviewer_5sH1 · 2026-04-03
> >
> > The author addressed most of my concerns in their rebuttal,

---

### Official Review · Reviewer_is1c · 2026-03-09

**Soundness:** 3
**Presentation:** 3
**Significance:** 3
**Originality:** 2
**Overall Recommendation:** 5
**Confidence:** 4

**Summary:**

The paper compares and revisits cross-entropy classification (CE) and supervised contrastive learning (SCL), to highlight their similarities and to favor their convergence towards neural collapse (NC).

**Compliance With Llm Reviewing Policy:**

Affirmed.

**Final Justification:**

Rebutal is convincing. I keep the 'accept' because this paper is solid and expected to have a high impact, probably not exceptional.

**Key Questions For Authors:**

See strengths and weaknesses.
The 'accept' is conditioned to a clear reply regarding the concerns raised among 'weaknesses', in particular the link with associative embedding methods.

**Limitations:**

Not an issue.

**Strengths And Weaknesses:**

Strengths:
•	By comparing CE and SCL, the paper offers a unified vision of the most popular supervised classification methods. It also proposes some meaningful variants, by contrasting with respect to instances instead of class prototypes (see NTCE), and by avoiding intra-class contrasts (see NONL).
•	The paper is reasonably well written, even if disentangling the CE/SCL comparison and the NC discussion would facilitate the reading (see weaknesses below).
•	Experimental validation, whilst limited, is reasonably convincing.


Weaknesses:
•	At the send of Section 3, and on page 6, first column, linear probing weaknesses are presented. But not all methods using supervised contrastive learning rely on linear probing. Positioning the work with respect to the literature related to associative embedding (e.g. for instance segmentation, or category or team differentiation) would be welcome.
•	At the end of Section 3, the authors note that favoring NC with conventional CE requires complex regularization and hyperparameters tuning. They point out the benefit arising from the normalized softmax loss in Equation (3). What about previous studies maintaining *by design* the weight vectors on the hypersphere, e.g. using layer rotation as in ‘Layer rotation: a surprisingly simple indicator of generalization in deep networks?’ at https://deep-phenomena.org/
•	Section 4 mixes the CE/SCL comparative study, with the discussion about Neural collapse capabilities. The text would gain in clarity if those two components are distinguished. Hence, IMHO, the discussion about NC would deserve a specific section, coming after the comparative presentation of CE/SCL variants.

Minor: Please add an appropriate reference or the number of the corresponding section (or equation number) when referring to the methods in the tables reporting experimental results.

---

> ### Author Rebuttal · Authors · 2026-03-29
>
> We thank Reviewer is1c for the positive evaluation and for recognizing the unified vision bridging CE and SCL, the meaningful variants NTCE and NONL, and the convincing experimental validation. We also thank the reviewer for pointing to conceptually related literature. Below we discuss each point and welcome further discussion.
>
> ## W1: Positioning relative to associative embedding
>
>  The reviewer observes an important connection. Indeed, in associative embedding methods (Newell et al., NeurIPS 2017; Neven et al., 2019; De Brabandere et al., 2017) embeddings are learned by contrasting in-group positives to out-of-group negatives and are applied to tasks where linear probing is not needed. While our work focuses on the principled SCL pipeline for classification, where linear probing is standard practice and where NC theory provides formal optimality guarantees, **this conceptual link is meaningful and worth making explicit**.
>
> Our work connects to this literature concretely. While eliminating linear probing is not relevant to associative methods (which already skip it), **our Theorem 4.2 is**: it shows that optimizing per-instance alignment and uniformity (as in SCL) yields the same optima as optimizing with class-mean prototypes (as associative methods do). This **gives theoretical room to apply SCL in associative method applications**.
>
> However, associative methods operate in unconstrained Euclidean space where precisely the radial degree-of-freedom problem we identify prevents convergence to optimal geometry. Our hyperspherical framework addresses this: normalizing embeddings to the unit sphere eliminates scale ambiguity and provides geometric guarantees that associative methods lack. Our results suggest one could replace associative losses with SCL on the hypersphere, achieving the same optima more easily thanks to the benign loss landscape (Yaras et al. 2022).
>
> Beyond SCL, NTCE and NONL open room for principled and more efficient CE-style training that uses learnable group embeddings (the classifier weights). We will expand this discussion in the paper which leaves room for future research on associative embedding methods.
>
> ## W2: Alternative approaches to constraining radial freedom
>
>  This is a very interesting question and we will update Related Work to present the complete landscape. There are several approaches beyond normalized softmax: Riemannian optimization on the Stiefel manifold (Huang et al., AAAI 2018; Bansal et al., NeurIPS 2018), weight normalization (Salimans & Kingma, NeurIPS 2016), spectral normalization (Miyato et al., ICLR 2018), and layer rotation (Carbonnelle & De Vleeschouwer, 2019).
>
> The critical observation is that all these methods are applied on top of standard CE, inheriting its fundamental limitations. Stiefel optimization constrains weight vectors to be mutually orthogonal (pairwise cosine = 0) by definition, whereas the simplex ETF that NC requires has pairwise cosine of $-1/(K-1)$, thus  constituting **a distinct geometric structure**. Our normalized softmax can be understood as optimization on the oblique manifold (product of spheres), which Yaras et al. (2022) prove has a benign strict-saddle landscape specifically for NC. **Weight and spectral normalization control magnitudes but still operate with unnormalized features and learnable biases**. Layer rotation is a useful diagnostic but does not modify the training objective. In all cases, applying these constraints to CE still leaves: (i) additional regularization hyperparameters to eliminate radial freedom, (ii) no temperature $\tau$ to control the similarity distribution, and (iii) CE's fundamental loss-level limitations, ie small effective negative set and alignment-uniformity coupling, that no geometric constraint alone can resolve.
>
> **Normalized softmax addresses the constraint** side cleanly by utilizing per-vector normalization on the right manifold, zero bias, temperature control, while **NTCE and NONL address the loss-level problems** that persist even under perfect geometric constraints.
>
> ## W3: Separating CE/SCL comparison from NC discussion
>
> We agree this would improve clarity. We propose: Section 4 introducing NTCE, NONL, and the fixed-prototype classifier with their motivations; Section 5 dedicated to Neural Collapse analysis presenting Theorems 4.1 and 4.2, showing how both objectives converge to optimal geometry on the hypersphere. The reader first understands what the methods are, then why they achieve optimal geometry.
>
> **Minor.** Added equation/section references for all methods in the experimental tables.

---

> > ### Author Rebuttal · Reviewer_is1c · 2026-04-01
> >
> > Thanks for the complete and convincing reply.

---

### Official Review · Reviewer_WY8e · 2026-03-13

**Soundness:** 2
**Presentation:** 3
**Significance:** 2
**Originality:** 2
**Overall Recommendation:** 4
**Confidence:** 4

**Summary:**

Three key observations drive the motivation of this work: the potential of Neural Collapse geometric properties as an inductive bias for improved generalization, the structural similarity between cross-entropy loss and supervised contrastive learning, and the empirical finding that supervised contrastive learning inherently induces a simplex ETF structure in the learned feature space even in the absence of linear probing. Grounded in these observations, the authors introduce two sample-level losses (NTCE, NONL) derived from cross-entropy, both operating within mini-batches. Empirical evaluations and theoretical analysis validate the effectiveness of the proposed approach, demonstrating its ability to induce Neural Collapse more effectively.

**Compliance With Llm Reviewing Policy:**

Affirmed.

**Final Justification:**

As my concerns were fully resolved by the rebuttal, I would like to raise my score to 4. weak accept., and I plan to follow the ongoing discussion among reviewers during the remaining period before making my final decision.

(Update my final justification) Upon reviewing the comments from other reviewers, it appears that concerns regarding the novelty of the theorems have not yet been fully resolved. As this is to some extent aligned with my official review, and given that it bears directly on the contribution of the paper, I will maintain my current score.

**Key Questions For Authors:**

Please refer to the Weaknesses

**Limitations:**

yes

**Strengths And Weaknesses:**

### Strengths

* The paper provides rigorous proofs that the proposed objectives (NTCE, NONL) and NormFace all achieve Neural Collapse as their global optimum under the balanced UFM/LPM setting.

* The proposed methods are validated across four benchmarks including ImageNet-1K, consistently surpassing cross-entropy in both accuracy and NC metrics. In addition, the benefits extend beyond standard classification to transfer learning, long-tailed recognition, and robustness, demonstrating broad practical utility.

* The paper identifies two underappreciated computational bottlenecks: i) a small effective negative set in classifier learning and ii) the alignment-uniformity coupling in contrastive objectives. In addition, this work addresses them with principled algorithmic modifications.

### Weaknesses

**W1.** Except in the special cases of transfer learning and class imbalance, the improvements in model performance remain marginal across the remaining experimental settings. Therefore, the generalizability of the proposed NTCE and NONL appears to be overestimated.

**W2.** In standard image classification benchmarks (Tables 1 and 2), SCL+FP achieves the most competitive performance among the proposed methods, although the gains remain marginal on benchmarks other than IN-100. However, SCL+FP lacks novelty, as it is essentially a combination of previously established findings from the Simplex ETF of SCL in [R1,R2] and NC4-NCC in Neural Collapse, making it difficult to propose a novel method as an independent contribution.

**W3.** [lines 090-097] Furthermore, since [R1] and [R2] have already presented more general theorems under the condition that all last-layer features and class weight vectors are norm-bounded ($||z|| \leq \rho_{\mathcal{Z}}, ||w|| \leq r_{\mathcal{W}}$), C3 appears redundant with respect to these prior works and seems to be merely a special case where the radius is 1 ($||z|| = \rho_{\mathcal{Z}} = 1, ||w|| = r_{\mathcal{W}} = 1$). Could you elaborate on the distinctions and advantages of the proposed theorem compared to [R1] and [R2]?

**W4.** According to prior work on the effect of collapsed representations on transfer learning, a trade-off exists between neural collapse and transfer learning performance, where a lower degree of neural collapse leads to better transfer learning performance ([R2]: *Collapsed representations cannot distinguish fine-grained details within*, [R3]: *Representations with higher class separation obtain higher accuracy on the original task, but their features are less useful for downstream tasks*, [R4]: *The encoder is trained to prevent NC and encourage transferable representations for OOD generalization*). However, contrary to these prior findings, the proposed methods consistently outperform baselines across all datasets except DTD in Table 4. In my opinion, this discrepancy can be attributed to the following two reasons.

i) In this transfer learning experiment, the encoder of the ImageNet-pretrained model is frozen, and only the classifier is fine-tuned on the target dataset. Consequently, the distribution of last-layer features remains unchanged. Under this setting, do NTCE and NONL still exhibit higher NC metric values than the baselines? Could the authors provide additional NC metric values measured after fine-tuning on the target datasets?

ii) Since the number of classes in ImageNet is larger than that of the target datasets used for fine-tuning, the trade-off may not manifest severely. As illustrated in the coarse-to-fine transfer setting of [R2], when a model pretrained on coarse-grained classes (e.g., vehicle in CIFAR-10) is transferred to fine-grained classes (e.g., car, truck in CIFAR-10), classifying inputs into fine-grained classes becomes difficult due to collapsed representations. Could the authors provide additional transfer learning results on a dataset with more classes than ImageNet?

If the proposed methods exhibit higher NC metric values in the additional experiment of i) and outperform baselines in the additional experiment of ii), this would lead to a conclusion contradicting prior work, namely that a higher degree of neural collapse yields more transferable representations. Could the authors elaborate on why the proposed methods produce such contradictory results?

---

> ### Author Rebuttal · Authors · 2026-03-29
>
> We thank Reviewer WY8e for the careful review. We address each concern below.
>
> ## W1: Performance improvements
>  On **ImageNet-1K**, NTCE achieves **+1.3pp** and NONL **+0.9pp** over CE, with **consistent gains** across all datasets  by **just modifying the loss**. Our +0.3pp over NormFace on IN-1K is **~1.5K additional correct classifications**. Combined with transfer, class imbalance gains, and faster convergence, this is a clear practical benefit.
>
> ## W2–W3: SCL+FP novelty and C3
>
>  We note that **NTCE and NONL, not SCL+FP, are our most competitive methods** (+1.6pp and +1.4pp over SCL+FP on IN-1K). Results are in separate tables because the two families use different pipelines (heavy augmentation for SCL, etc.) and thus, in the paper, we avoid comparisons between method families.
>
> Below we show two contributions over prior work: (i) in theory, Theorem 4.2 proves a classifier is implicit in SCL, establishing NC3/NC4 for the first time; (ii) in practice, SCL+FP classifies directly on the normalized optimization space, departing from standard SCL which discards it and probes unnormalized embeddings with unnormalized CE (weight magnitudes and biases).
>
> **C3:** SCL has no classifier weights in its formulation. Prior theorems therefore establish NC1+NC2 for SCL minimizers but **cannot address NC3 (classifier-feature alignment) or NC4 (zero bias)** which are not even expressible without a classifier. Theorem 4.2 resolves this by proving that **a classifier is already implicit in SCL**: the class-mean prototypes *are* the classifier weights, enabling NC3 and NC4.
>
> This is not a trivial consequence of NC1+NC2. Prior results characterize what SCL solutions *look like* (collapsed ETF); Theorem 4.2 proves **what SCL optimizes for** (prototype classification). Concretely, it establishes that **class-mean prototypes do not serve as post-hoc classifier weights but are the solution the SCL objective drives toward**, making their direct use principled rather than heuristic. This matters in practice: exact NC is never reached, so the minimizer-set equivalence, ie guaranteeing the objective drives toward prototype-optimal classifiers *throughout training*, is why FP works at ~95% NC (Table 3), not only at exact convergence.
>
> **Practical novelty**: The linear probing phase of SCL discards the projection head, a convention inherited from SSL, and trains on *unnormalized encoder representations with unconstrained weights and biases*. **SCL+FP uses the normalized representations from the projection head (optimization space), adding no magnitude and bias**. This rests on two contributions: (i) Theorem 4.2 proves the optimal classifier is already on the optimization space, and (ii) our normalized classifier framework which enables classification on the normalized space.
>
> ## W4: Transfer results are grounded in the literature
>
> **Transfer Protocol (i).** We do **not** claim our methods achieve NC on the target transfer task. We freeze the encoder and train a linear probe with **CE for all methods identically**, thus NC metrics on the target reflect CE training and not our objectives. Our claim is narrower: representations reaching NC during pretraining transfer better. Tables 4–6 validate this.
>
> **On coarse-to-fine transfer ([R2]) and (ii).** In the standard transfer setting we evaluate, knowledge transfers from large-scale to smaller downstream tasks (Khosla et al. 2020; Chen et al. 2020), not from coarse to fine-grained sub-classes. Transfer to a dataset with more classes than ImageNet-1K is outside our computational reach.
>
> We thank the reviewer for this connection to the **different coarse→fine transfer setting** where [R2] argues collapsed representations cannot distinguish fine-grained details within a coarse class. **In the revised paper we will acknowledge and discuss the possible limitation of NC methods under this setup**.
>
> **On the cited literature.** From context we identify [R2] as Ma et al. (ICLR 2023), [R3] as Kornblith et al. (NeurIPS 2021), [R4] as Harun et al. (ICML 2025). The cited works do not contradict our findings:
>
> 1.  The broader NC literature establishes that collapsed, maximally separated representations improve generalization, robustness, and transfer (in the standard transfer setting)[1-9].
>
> 2.  [R4] studies *intermediate-layer* NC and transfer. Their method **explicitly enforces NC at the classification layer**, the layer we operate on, achieving strong transfer and **validating NC at the final layer is compatible with strong transfer**.
>
> 3.  [R3] reduces within-class variance without principled inter-class structure, which harms transfer. NC jointly minimizes within-class variance and maximizes between-class separation (ETF).
>
>
> [1] Papyan et al. 2020 [2] Bartlett et al. 2017 [3] Neyshabur et al. 2018 [4] Fawzi et al. 2016 [5] Ding et al. 2020 [6] Galanti et al. 2021 [7] Khosla et al. 2020 [8] Soudry et al. 2018 [9] Hein & Andriushchenko 2017

---

> > ### Author Rebuttal · Reviewer_WY8e · 2026-04-03
> >
> > We sincerely apologize for accidentally omitting the reference list in our review. Nevertheless, we are truly impressed and grateful that the authors were able to identify and respond to the intended references based solely on the provided context. We also thank the authors for their detailed and thoughtful response. To clarify our intentions: we are fully aware of the authors' contributions — namely, that the class-mean prototype in SCL already forms a simplex ETF, and that this insight was leveraged to propose the novel method NTCE NONL. Through our review, we aimed to raise discussions on the novelty relative to prior work (W2–3) and the potential side effects of NC in transfer learning settings (W4), with the hope that these discussions would help sharpen and strengthen the authors' contributions. Furthermore, despite our misunderstanding of the transfer protocol — specifically, that all methods share a frozen pre-trained encoder and each method is applied only during fine-tuning on the target task — the authors kindly clarified this point. With this concern regarding transfer learning now resolved, we believe the marginal improvement noted in W1 can also be considered negligible.
> >
> > In summary, I would like to raise my score to 4. weak accept, and I plan to follow the ongoing discussion among reviewers during the remaining period before making my final decision.
> >
> >
> >
> > (Here are the references from the review, albeit late.)
> >
> > [R1] Graf et al., Dissecting supervised constrastive learning, ICML'2021
> >
> > [R2] Chen et al., Perfectly balanced: Improving transfer and robustness of supervised contrastive learning, ICML'2022
> >
> > [R3] Kornblith et al., Why do better loss functions lead to less transferable features?, NeurIPS'2021
> >
> > [R4] Harun et al., Controlling Neural Collapse Enhances Out-of-Distribution Detection and Transfer Learning, ICML'2025

---

### Decision · Program_Chairs · 2026-04-30

**Decision:**

Accept (regular)

**Comment:**

This paper focuses on the practical gap between supervised learning and the optimality of Neural Collapse (NC) geometry by unifying normalized softmax classification and supervised contrastive learning as prototype-based learning on a unit hypersphere. It introduces two novel losses, NTCE and NONL, which expand negative samples and decouple intra-class alignment from inter-class repulsion to speed up NC convergence. The paper theoretically proves that supervised contrastive learning already learns optimal class prototypes, eliminating the need for linear probing. Experiments on standard vision benchmarks are conducted to show the effectiveness of the methods to achieve NC and the comparison with baseline methods, e.g. ETF + DR.

Reviewers have pointed out several concerns: most notably, performance gains are marginal across most standard settings; some theoretical results seem largely redundant with respect to prior works; and the empirical evaluation lacks statistical significance tests. After rebuttal, three reviewers acknowledged their concerns fully resolved. One reviewer had follow-up questions and acknowledged that the authors' further response addressed them. After reading the paper and all the comments, the AC agrees that the paper provides a clean unifying framework for normalized classification and supervised contrastive learning on the hypersphere and may deliver practically useful benefits.